

# Precursors and formation of secondary organic aerosols from wildfires in the Euro-Mediterranean region

Marwa Majdi[1-2], Karine Sartelet[1], Grazia Maria Lanzafame[3], Florian Couvidat[3], Youngseob Kim[1], Mounir Chrit[1], and Solene Turquety[2]

[1]CEREA: joint laboratory École des Ponts ParisTech EDF R&D, Université Paris-Est, 77455 Champs sur Marne, France
[2]Laboratoire de Météorologie Dynamique (LMD)-IPSL, Sorbonne Université, CNRS UMR 8539, Ecole Polytechnique, Paris, France.
[3]INERIS: Institut National de l'Environnement Industriel et des Risques, Verneuil en Halatte, France

*Correspondence to:* Marwa Majdi (marwa.majdi@enpc.fr)

**Abstract.** This work aims at quantifying the relative contribution of secondary organic aerosol (SOA) precursors emitted by wildfires to organic aerosol (OA) formation, during summer 2007 over the Euro-Mediterranean region, where intense wildfires occurred. A new SOA formation mechanism, $H^2O_{aro}$, including recently identified aromatic volatile organic compounds (VOCs) emitted from wildfires is developed based on smog chamber experiment measurements, under low and high-$NO_x$

regimes. The aromatic VOCs included in the mechanism are toluene, xylene, benzene, phenol, cresol, catechol, furan, naphthalene, methylnaphthalene, syringol, guaiacol and structurally assigned and unassigned compounds with at least 6 carbon atoms per molecule (USC>6). This mechanism $H^2O_{aro}$ is an extension of the $H^2O$ (Hydrophilic/Hydrophobic organic) aerosol mechanism: the oxidation of the precursor forms surrogate species with specific thermodynamic properties (volatility, oxidation degree, affinity to water). The SOA concentrations over the Euro-Mediterranean region in summer 2007 are simulated

using the chemistry transport model (CTM) Polair3D of the air-quality plateform Polyphemus, where the mechanism $H^2O_{aro}$ was implemented. To estimate the relative contribution of the aromatic VOCs, intermediate, semi and low volatile organic compounds (I/S/L-VOCs) to wildfires OA concentrations, different estimations of the gaseous I/S/L-VOC emissions (from primary organic aerosol (POA) using a factor of 1.5 or from non-methanic organic gas (NMOG) using a factor of 0.36) and their ageing (one-step oxidation vs multi-generational oxidation), are also tested in the CTM.

Most of the particle organic aerosol (OA) concentrations are formed from I/S/L-VOCs. In average during the summer 2007 and over the Euro-Mediterranean domain, they are about 10 times higher than the OA concentrations formed from VOCs. However, locally, the OA concentrations formed from VOCs can represent up to 30% of the OA concentrations from biomass burning. Amongst the VOCs, the main contributors to SOA formation are phenol, benzene and catechol (47%), USC>6 compounds (23%), and toluene and xylene (12%). Sensitivity studies of the influence of the VOCs and the I/S/L-VOCs emissions

and chemical ageing mechanisms on $PM_{2.5}$ concentrations show that surface $PM_{2.5}$ concentrations are more sensitive to the parameterization used for gaseous I/S/L-VOCs emissions than for ageing. Estimating the gaseous I/S/L-VOCs emissions from POA or from NMOG has a high impact on local surface $PM_{2.5}$ concentrations (reaching -30% in Balkans, -8 to -16% in the fire plume and +8 to +16% in Greece). Considering the VOC emissions results in a moderate increase of $PM_{2.5}$ concentrations





mainly in Balkans (up to 24%) and in the fire plume (+10%).

# 1   Introduction

Atmospheric particulate matter (PM) has a strong impact on human health (Pope et al., 2002; Naeher et al., 2006; Johnston
et al., 2012), climate (Pilinis et al., 1995; Bond et al., 2013) and visibility (Eldering and Cass, 1996; Hand et al., 2007). Chemistry transport models (CTMs) play an important role in simulating the formation of these particles and their concentrations.
PM is composed of different compounds: organics, inorganics, dust, black carbon (Jimenez et al., 2009).

Organic aerosols (OA) are classified either as primary (POA) or as secondary aerosols (SOA). POA are directly emitted into
the atmosphere, whereas SOA are formed by gas-particle conversion of oxidation products of precursors. OA can be classified based on their saturation concentrations ($C^*$): volatile organic compounds (VOCs) (with $C^* > 10^6 \mu g.m^{-3}$), intermediate
organic compounds (I-VOCs) (with $10^4 < C^* < 10^6 \mu g.m^{-3}$), semi-volatile organic compounds (S-VOCs) (with $0.1 < C^* < 10^4 \mu g.m^{-3}$) and Low-volatility organic compounds (LVOC) (with $C^* < 0.1 \ \mu g.m^{-3}$) (Lipsky and Robinson, 2006; Grieshop
et al., 2009). As SOA, POA may be composed of components of different volatilities such as S-VOCs, L-VOCs which may partition between the gas and particle phases (Robinson et al., 2007). Depending on the ambient concentrations, some components
only exist in the gas phase (e.g. I-VOCs). In the following, $OA_{tot}$ denotes the sum of gas and particle phase concentrations.
$POA_{tot}$ originate mostly from anthropogenic (e.g. traffic, industry) sources and from biomass burning, which is considered as
one of the major sources of PM (Bian et al., 2017), with contributions from both anthropogenic (e.g. residential heating) as
well as natural sources such as wildfires.

Wildfire is one of the largest sources of primary carbonaceous aerosols globally. It is also an important source of trace gases
including organic vapors which themselves can serve as precursors of SOA (Akagi et al., 2011; Stockwell et al., 2015). SOA
from wildfires may contribute significantly to organic aerosol loading in the atmosphere (Konovalov et al., 2015). However, the
concentration of SOA is highly uncertain because of the complexities of physical and chemical evolution of wildfire plumes
(Bian et al., 2017). Although several modeling studies have examined SOA formation from VOCs released from biomass
burning (Marson et al., 2006; Alvarado and Prinn, 2009; Alvarado et al., 2015), the compounds that act as precursors of SOA
are still not well understood. Considering only traditional SOA precursors (mainly toluene, xylene, benzene and naphthalene
(Pye et al., 2017)) in SOA models leads to a substantial underestimation of SOA concentrations (Dawson et al., 2016; Bian
et al., 2017). This can probably partly be explained by the limited knowledge about SOA precursors. Recently, aromatic VOCs
(namely toluene, xylene, benzene, phenol, cresol, catechol, furan, guaiacol, syringol, naphthalene, methylnaphthalene) were
identified as the major SOA precursors emitted by biomass burning (Akagi et al., 2011; Stockwell et al., 2015; Bruns et al.,
2016). To develop mechanisms of SOA formation from these aromatic compounds, many laboratory studies have investigated
the gas-phase oxidation of VOCs (mainly initiated by reactions with hydroxyl radical (OH)) (Calvert et al., 2002; Atinkson
and Arey, 2003; Chhabra et al., 2011; Nakao et al., 2011; Yee et al., 2013) and SOA yields have been measured under various
conditions (Odum et al., 1996a; Ng et al., 2007): low-$NO_x$ regime where the concentrations of $NO_x$ are low and the produc-





tion of ozone and oxidants is mainly governed by the $NO_x$ levels, and high-$NO_x$ regime where the production of ozone and oxidants is controlled by the VOC levels (Sillman et al., 1990; Kleinman, 1994). Odum et al. (1996a) model SOA formation by a gas/particle partitioning absorption scheme (Pankow, 1994) using data from smog chamber experiments. In CTMs, the SOA formation may be represented using different approaches mostly based on data from smog chamber experiments: the

two lumped product approach, which uses an empirical representation of SOA formation (Odum et al., 1996a; Schell et al., 2001), the molecular or surrogate approach (Pun et al., 2006; Bessagnet et al., 2008; Carlton et al., 2010; Couvidat et al., 2012; Chrit et al., 2017), which represents the formation of SOA using surrogate molecules with associated physico-chemical properties; the Volatility Basis Set (VBS) approach (Donahue et al., 2006), in which surrogates are associated to classes of different volatilities. The ageing (oxidation by OH) of each surrogate may lead to the formation of surrogates of lower volatility classes

through the competition of two processes: fragmentation and functionalization.

SOA formation mechanisms may rely not only on smog chamber experiments, but also on explicit chemical mechanisms when experimental data are not available. Examples of such mechanisms are the master chemical mechanisms (MCM) (Saunders et al., 1997) or the generator for explicit chemistry and kinetics of organics in the atmosphere (GECKO-A) (Aumont et al., 2005).

Recent studies take into account not only the oxidation of selected VOCs but also gaseous I/S/L-VOCs emitted by biomass burning to model SOA formation (Koo et al., 2014; Konovalov et al., 2015; Giancarlo et al., 2017). Majdi et al. (2018) show that near fire regions and during the summer 2007, 52% to 87% of the $PM_{2.5}$ concentrations are organic aerosol that are mainly composed of primary and secondary I/S/L-VOCs (62 to 84%). They highlight that neglecting primary gaseous I/S/L-VOCs emissions from wildfires tends to lessen the surface $PM_{2.5}$ concentrations (-30%). Since ignoring primary gaseous I/S/L-VOCs

emissions biases model predictions of SOA production, several studies based on smog chamber data aim at estimating them (Yokelson et al., 2013; Jathar et al., 2014, 2017). The primary gaseous I/S/L-VOCs emitted by biomass burning are usually calculated using the emissions of POA (Couvidat et al., 2012; Koo et al., 2014) because a part of these I/S/L-VOCs may correspond to POA due to the gas-to-particle partitioning. However, these gaseous I/S/L-VOC emissions may also correspond to an unspeciated fraction of non-methane organic gas (NMOG) (Jathar et al., 2014, 2017). Jathar et al. (2014) estimate that

about 20% of the total NMOG emitted from biomass burning is assumed to be I/S/L-VOCs in the gas phase, while Yokelson et al. (2013) estimate that as much as 35% to 64% of NMOG is I/S/L-VOCs in the gas phase.

Although primary gaseous I/S/L-VOCs are not considered or misclassified in emissions inventories, their contribution to the SOA budget may be substantial, despite being a small fraction of the overall organic gas emissions (Koo et al., 2014; Konovalov et al., 2015; Giancarlo et al., 2017). The gaseous I/S/L-VOCs are usually classified according to their volatilities

(Couvidat et al., 2012; May et al., 2013) by taking into account the variation of their average oxidation state (Koo et al., 2014). Different parameterizations have been used to simulate the ageing of gaseous I/S/L-VOCs emitted by the biomass burning: a simple one-step oxidation scheme (Couvidat et al., 2012), or a multi-generational oxidation scheme taking into account simultaneously functionalization and fragmentation at each step (Koo et al., 2014; Giancarlo et al., 2017).




The objective of this work is to quantify the contribution of recently identified SOA precursors from wildfires (toluene, xylene, guaiacol, syringol, benzene, phenol, catechol, cresol, furan, naphthalene, methylnaphthalene and USC>6 compounds). To that end, a SOA formation mechanism is developed for those precursors, based on smog chamber experiments under low and high-$NO_x$ conditions. This study aims also to quantify the relative contribution of VOCs and I/S/L-VOCs on OA formation.

The OA concentrations are simulated using the CTM Polair3D of the Polyphemus modeling air-quality platform.

This study focuses on two severe fire events that occured during the summer of 2007 over the Euro-Mediterranean area. Majdi et al. (2018) show a large contribution of wildfires (reaching $\sim 90\%$) mainly in Greece (24–30 August 2007) and in Balkan (20–31 July 2007, 24–30 August 2007). Through comparisons to both ground based and satellite remote sensing (MODIS) observations, a general good performance for surface modeled $PM_{2.5}$ with a clear improvement of $PM_{2.5}$ is found when

including fire emissions.

This paper is structured as follows. Section 2 details the SOA formation mechanisms from VOCs and I/S/L-VOCs. Then, section 3 describes the model and the simulation set-up during summer 2007. The main $OA_{tot}$ precursors (VOCs, gaseous I/S/L-VOCs) emitted from wildfires, their emission factors and their emissions are detailed in section 4. Section 5 presents the sensitivity simulations performed to understand the relative impact of VOCs and I/S/L-VOCs on OA formation.

## 2 SOA formation from VOCs and I/S/L-VOCs

### 2.1 SOA formation from VOC oxidation

This section presents a new SOA formation mechanism $H^2O_{aro}$ developed to represent the SOA formation from the main VOCs that are estimated to be SOA precursors (phenol, cresol, catechol, benzene, furan, guaiacol, syringol, naphthalene,

methylnaphthalene). The new mechanism ($H^2O_{aro}$) is an extension of hydrophilic/hydrophobic organic ($H^2O$) SOA mechanism. Laboratory chamber studies provide the fundamental data that are used to parameterize the atmospheric SOA formation under low/high-$NO_x$ conditions. All the experiments used in this paper were conducted under dry conditions with a relative humidity (RH) lower than 10% and a temperature ranging between 292 and 300 K .

For each VOC, precursor of SOA, and each chamber experiment, the SOA mass yield (Y) is defined as the fraction of the

25 reactive organic gas (ROG) that is converted to SOA. The relationship between the yield and the measured organic aerosol mass concentration (i.e. formed SOA) $M_0$ (Odum et al., 1996a) is:

$$Y = \sum_{i=1}^{n} \frac{\alpha_i K_{p,i}.M_0}{(1 + K_{p,i}.M_0)} \tag{1}$$

where $\alpha_i$ is the molar stoechiometric coefficient of the product (surrogate) $i$, and $K_{p,i}$ is its gas-particle partitioning equilibrium constant.

The chamber experimental results are analyzed according to the absorption gas-particle partitioning model developed by Pankow (1994) and Odum et al. (1996a). For each VOC, the experimental results (Y, $M_0$) are fitted (with the least mean square method) either with one product model or two products model by plotting the Odum curve. The stoechiometric coeffi-





cients of SOA products, their saturation vapor pressures and their partitioning gas-particle constants are determined from the experimental results and the Odum curve. Then candidates for SOA surrogates formed by the VOC oxidation are estimated from the literature. For each candidate, the saturation vapor pressure and the partitioning constant are estimated from an empirical method called "the group contribution method" proposed by SIMPOL.1 (Pankow and Asher, 2008). These parameters

are used to choose the SOA surrogates amongst the candidates: the SOA surrogates are chosen so that their saturation pressure and partitioning constant are the closest to the ones determined experimentally from the Odum plot.

### 2.1.1 Oxidation of phenol and catechol

Under low-$NO_x$ conditions, the chamber experiments of Yee et al. (2013), Chhabra et al. (2011) and Nakao et al. (2011) are

used to model the SOA formation from phenol oxidation.

In their studies, and in agreement with the explicit chemical mechanism MCM.V3.3.1, catechol (CAT) is the dominant product of the first oxidation step of phenol. Therefore, catechol is assumed to be the main intermediary leading to SOA formation from OH oxidation of phenol following reaction R1.

$$PHEN + OH \xrightarrow{k_1} 0.75\, CAT \tag{R1}$$

where the kinetic constant $k_1 = 4.7\ 10^{-13}\ \exp(1220/T)$ molecule$^{-1}$.cm$^3$.s$^{-1}$ and the stoechiometric coefficient of catechol are given by MCM.v3.3.1. SOA from phenol are produced essentially from the oxidation of catechol, which is mostly present in the gas phase ($K_p = 2.57$ m$^3$.g$^{-1}$). The yields of the SOA surrogates formed from the catechol oxidation by OH are estimated assuming that reaction (R1) holds and using the Odum approach with the results (yields and $M_0$) of the experiments conducted by Yee et al. (2013) and Chhabra et al. (2011) for phenol oxidation. The Odum approach (Odum et al., 1996a) is used here

with only one surrogate (one-product model) to estimate SOA formation parameters, as similar partitioning constants and stoechiometric coefficients are obtained with two surrogates. Figure 1 shows the plots of the SOA yields against the SOA concentrations $M_0$. The blue squares are yields from smog chamber experiments and the orange diamonds are yields estimated by the one-product model. The one-product model with a stoechiometric coefficient $\alpha_1$ of 0.28 and a vapor pressure of 4.59 $10^{-8}$ torr correctly reproduces the experimental data. Note that this stoechiometric coefficient (0.28) is similar to the one

obtained using the experimental result of Nakao et al. (2011) for the OH oxidation of catechol (0.26). Yee et al. (2013) identified SOA products from phenol oxidation under low-$NO_x$ conditions. For each product proposed by Yee et al. (2013), vapor saturation pressures are calculated with SIMPOL.1 using "the group contribution method". The surrogate is chosen so that its estimated saturation vapor pressure corresponds to the experimental one estimated from the Odum curve. The product ACIDMAL ($C_6H_6O_5$, maleylacitic acid) is chosen as its theoretical vapor pressure ($5.76\ 10^{-8}$ torr) is the closest to

the experimental one ($4.59\ 10^{-8}$ torr). The Van Krevelen diagram in Chhabra et al. (2011) presents the properties of SOA from phenol oxidation in term of O/C and H/C ratios. According to the Van Krevelen diagram, the O/C and H/C ratios of SOA from phenol vary from 0.8 to 1 and between 1 and 1.5 respectively. This confirms that ACIDMAL is an acceptable SOA surrogate



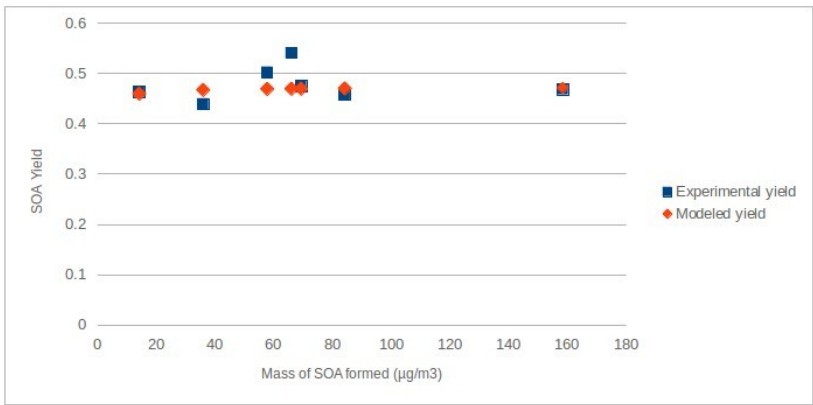

**Figure 1.** SOA yield from smog chamber experiments under low-NOx conditions (Yee et al., 2013; Chhabra et al., 2011; Nakao et al., 2011) and yield curve for phenol-OH reaction using one-product model.

for the OH oxidation of phenol (O/C =0.83 and H/C =1). Because of the lack of experimental data of phenol oxidation under high-$NO_x$, ACIDMAL is also used as high-$NO_x$ surrogate.

Finally, the oxidation of catechol is modeled following reaction R2.

$$CAT + OH \xrightarrow{k_2} 0.28 \ ACIDMAL \tag{R2}$$

where the kinetic constant $k_2 = 9.9 \ 10^{-10}$ molecule$^{-1}$.cm$^3$.s$^{-1}$ is taken from MCM.v3.3.1.

### 2.1.2 Oxidation of cresol

As detailed in the chemical mechanism MCM.v3.3.1, the OH oxidation of cresol (CRESp) leads to the formation of methyl-catechol (MCAT), which is the dominant product of the first oxidation step of cresol, presented in reaction R3.

$$CRESp + OH \xrightarrow{k_3} 0.73 \ MCAT \tag{R3}$$

where the kinetic constant $k_3 = 4.65 \ 10^{-10}$ molecule$^{-1}$.cm$^3$.s$^{-1}$ and the stoechiometric coefficient are from MCM.v3.3.1.

The oxidation of methylcatechol by OH leads to the formation of SOA, following a chemical mechanism detailed in Schwantes et al. (2017). Because of the lack of the experimental data under high-$NO_x$ conditions, we consider that cresol chemical mechanisms under low and high-$NO_x$ conditions are similar. Aerosol yields from the experiments of Nakao et al. (2011) under low-$NO_x$ conditions is used for the Odum approach. The one-product model is sufficiently accurate to reproduce correctly the data from smog chamber. Figure 2 plots the SOA yields against the SOA concentrations. A stoechiometric coefficient and a saturation vapor pressure 0.39 and 3.52 $10^{-6}$ torr respectively are found to fit accurately the experimental data. The oxidation mechanism of MCAT developed by Schwantes et al. (2017) presents the potential candidates of SOA surrogates. For each





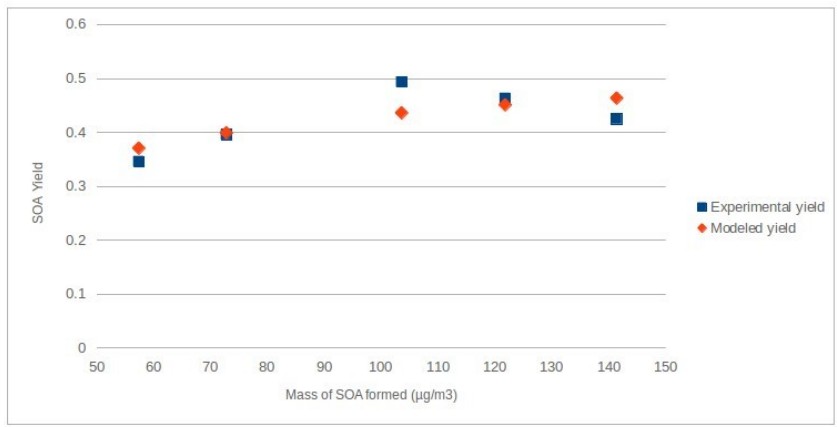

**Figure 2.** SOA yield data from smog chamber under low-NOx conditions (Nakao et al., 2011) and yield curve for cresol-OH reaction using one-product model.

candidate, the theoretical vapor saturation pressure is calculated using SIMPOL.1. DHMB ($C_7H_6O_4$, dihydroxymethylbenzo-quinone) has the closest vapor saturation pressure ($4.2 \ 10^{-6}$ torr) to the experimental vapor pressure calculated from the Odum plot ($3.52 \ 10^{-6}$ torr), and it is also close to the experimental pressure given in Schwantes et al. (2017) ($6.3 \ 10^{-6}$ torr).

Finally, the oxidation of methylcatechol is modeled following the reaction R4:

$$MCAT + OH \xrightarrow{k_4} 0.39 \, DHMB \tag{R4}$$

where the kinetic constant $k_4$= $2 \ 10^{-10}$molecule$^{-1}$.cm$^3$.s$^{-1}$ is from MCM.v3.3.1 and the stoechiometric coefficient of DHMB is deduced from the Odum plot.

Several studies focus also on the oxidation of cresol by $NO_3$ (Olariu et al., 2013; Grosjean, 1990). This oxidation may not contribute significantly to SOA formation, because the $NO_3$ oxidation products of cresol are highly volatiles.

## 2.1.3 Oxidation of benzene

According to MCM.v3.3.1, benzene (BENZ) reacts with OH to form phenol, as presented in reaction R5.

$$BENZ + OH \xrightarrow{k_5} 0.53 \, PHEN \tag{R5}$$

where $k_5$ = $2.3 \ 10^{-12}$ exp(-190/T) molecule$^{-1}$.cm$^3$.s$^{-1}$ is from MCM.v3.3.1. For the case of benzene, only the formation through the phenolic route is taken into account for simplification purposes. However, due to the high SOA yield of phenol and the high amount of phenol formed through benzene oxidation, the phenolic route should be one of the main pathway for SOA formation. By using the phenol SOA mechanism developed previously in section 2.1.1, the SOA yield through the phenolic of 0.28 is evaluated. This yield is within the range of SOA yields from benzene oxidation (between 0.22 and 0.33) reported by



Nakao et al. (2011) for low-NOx conditions. It confirms that phenol is probably the main intermediate for the formation of SOA.

### 2.1.4  Oxidation of furan

According to MCM.v3.3.1, furan (FUR) reacts with OH to form an unsaturated 1,4-dicarbonyl product (butendial (ButDial)),
following the reaction R6.

$$FUR + OH \xrightarrow[k_6 \ \ 3]{} 0.87 \ ButDial \tag{R6}$$

where $k_6$ = 4.19 $10^{-11}$ molecule$^{-1}$.cm$^3$.s$^{-1}$ is from MCM.v3.3.1.

According to MCM.v3.3.1, ButDial reacts with OH to form highly volatile products (not detailed here because they may not form SOA) and a radical (RADButenalCOO), as presented in the reaction R7:

$$ButDial + OH \xrightarrow[k_7]{} 0.83 \ RADButenalCOO \tag{R7}$$

where $k_7$ = 5.2 $10^{-11}$ molecule$^{-1}$.cm$^3$.s$^{-1}$ is from MCM.v3.3.1.

Under high-NO$_x$ conditions, according to MCM.v3.3.1, the oxidation of RADButenalCOO forms highly volatile products (glyoxal and maleic anhydrid), which are not considered here for SOA formation (reaction R8):

$$RADButenalCOO + NO \xrightarrow[k_8]{} \tag{R8}$$

where $k_8$ = 7.5 $10^{-12}$ exp(980/T) molecule$^{-1}$.cm$^3$.s$^{-1}$ is from MCM.v3.3.1.

Under low-NO$_x$ conditions, the oxidation of RADButenalCOO forms malealdehydic acid (ButenalCOOH) as shown in the reactions R9 and R10:

$$RADButenalCOO + HO_2 \xrightarrow[k_9]{} 0.15 \ ButenalCOOH \tag{R9}$$

$$RADButenalCOO + RO_2 \xrightarrow[k_{10}]{} 0.3 \ ButenalCOOH \tag{R10}$$

where $k_9$ = 5.2 $10^{-13}$ exp(980/T) molecule$^{-1}$.cm$^3$.s$^{-1}$ and $k_{10}$ = 1.$10^{-11}$ molecule$^{-1}$.cm$^3$.s$^{-1}$ are from MCM.v3.3.1.

ButenalCOOH is mostly in the gas phase ($K_p$=1.53 $10^{-5}$ m$^3$.g$^{-1}$), and not in the particle phase. However, according to GECKO-A, it may be oxidized by OH to form a radical (RADButenalCOOHCOO) following the reaction R11:

$$ButenalCOOH + OH \xrightarrow[k_{11}]{} 0.3 \ RADButenalCOOHCOO \tag{R11}$$

where $k_{11}$ = 2.12 $10^{-11}$ molecule$^{-1}$.cm$^3$.s$^{-1}$ is from GECKO-A. The radical RADButenCOOHCOO can react similarly to RADButenCOO under low-NO$_x$ conditions to form the diacid (Buten(COOH)2) as presented in the reactions R12 and R13.

$$RADButenalCOOHCOO + HO_2 \xrightarrow[k_9]{} 0.15 \ Butenal(COOH)_2 \tag{R12}$$





$$RADButenalCOOHCOO + RO_2 \xrightarrow{\quad k_{10} \quad} 0.3\ Butenal(COOH)_2 \tag{R13}$$

Note that the oxidation mechanism of furan presented in this section probably overestimates the SOA concentrations from the OH oxidation route, because several reactions such as ozonolysis and photolysis of both ButenalCOOH and Butenal(COOH)$_2$

5 are not considered. These reactions may lead to the loss of the main intermediary responsible of SOA formation (Butenal-COOH and Butenal(COOH)$_2$).

Furthermore, other routes may be more efficient at forming SOA from furan. Jiang et al. (2018) showed that NO$_x$ levels and relative humidity (RH) may significantly influence SOA formation from furan, with higher SOA concentrations at high-NO$_x$ levels and high humidity.

### 2.1.5 Oxidation of syringol and guaiacol

According to Lauraguais et al. (2014), the SOA formation mechanisms from methoxyphenols namely syringol and guaiacol, is split in two steps. The first step consists in reactions (R14 and R15) with the radical OH:

$$SYR + OH \xrightarrow{\quad k_{12} \quad} RADSYR \tag{R14}$$

$$GUAI + OH \xrightarrow{\quad k_{13} \quad} RADGUAI \tag{R15}$$

where $k_{12}$ = 9.63 $10^{-11}$ molecule$^{-1}$.cm$^3$.s$^{-1}$ and $k_{13}$ = 7.53 $10^{-11}$ molecule$^{-1}$.cm$^3$.s$^{-1}$ are given by Lauraguais et al. (2012) and Coeur-Tourneur et al. (2010a) respectively.

The parameterization is developed for syringol and guaiacol by considering low-NOx and high-NOx conditions based on SOA

yields reported by Chhabra et al. (2011); Yee et al. (2013) and Lauraguais et al. (2012); Yee et al. (2013) respectively. Generally this compound represents low-NO$_x$ oxidation products. In this first parameterization it is also used as high-NO$_x$ surrogate. Figure 3 shows the modeled Odum plots for syringol SOA formation under both low-NO$_x$ and high-NO$_x$ conditions. A one-product parameterization is sufficient to properly represent the experimental data for the two regimes. The same surrogate compound can be used for both regimes as similar partitioning constants are estimated. Among the compounds recognized as

syringol oxidation products, C$_8$H$_{10}$O$_5$ (PSYR) is the only product with a vapor saturation pressure, calculated with SIMPOL.1 (7.53 $10^{-6}$ torr), close to the experimental one estimated from the Odum plot (7.72 $10^{-6}$ torr). Stoechiometric coefficients of 0.57 and 0.36 are also estimated from the Odum curve under low and high-NO$_x$ conditions respectively.

The second reaction step for SOA formation is then represented with the following reactions R16, R17 and R18:

$$RADSYR + HO_2 \xrightarrow{\quad k_{14} \quad} 0.57\ PSYR \tag{R16}$$



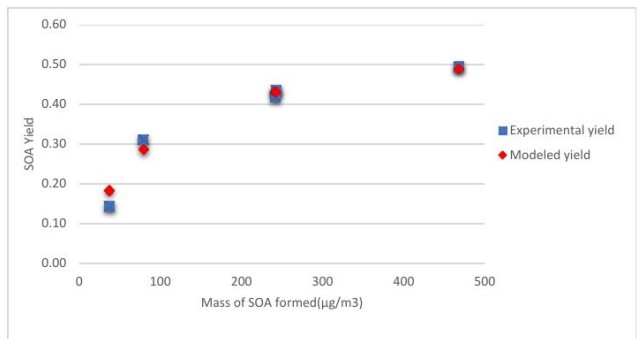 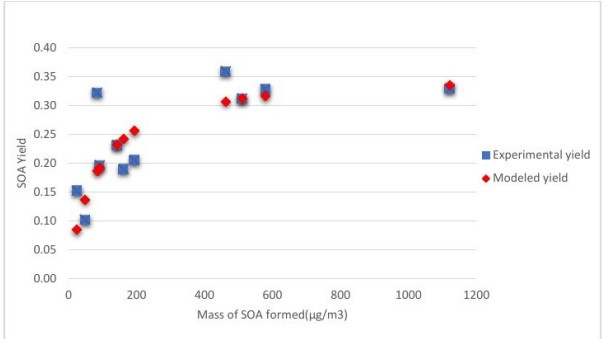

**Figure 3.** SOA experimental and modeled yield data from smog chamber for syringol under low-NO$_x$ conditions (left panel) (experimental data from Chhabra et al. (2011); Yee et al. (2013) and under high-NO$_x$ conditions (right panel) (experimental data from Yee et al. (2013); Lauraguais et al. (2012))

$$RADSYR + NO \xrightarrow{k_{15}} 0.36\ PSYR \tag{R17}$$

$$RADSYR + NO_3 \xrightarrow{k_{16}} 0.36\ PSYR \tag{R18}$$

where $k_{14}$ = 2.91 $10^{-13}$ exp(1300/T) molecule$^{-1}$.cm$^3$.s$^{-1}$, $k_{15}$ = 2.70 $10^{-13}$ exp(360/T) molecule$^{-1}$.cm$^3$.s$^{-1}$ and $k_{16}$ = 2.30 $10^{-12}$ molecule$^{-1}$.cm$^3$.s$^{-1}$ are from MCM.V3.3.1.

Similarly, for guaiacol, the two NO$_x$ regimes are distinguished. One surrogate compound is used for the high-NO$_x$ and the low-NO$_x$ parameterizations. Odum plots are presented in Figure 4. The surrogate compound chosen to represent SOA

formation in both conditions is C$_7$H$_{10}$O$_5$ (GHDPerox), a hydroperoxide proposed as an oxidation product for guaiacol in Yee et al. (2013). It was chosen because the calculated saturation vapor pressure with SIMPOL.1 (1.05 $10^{-6}$ torr) is close to the one estimated by the Odum method (6.01 $10^{-7}$ torr). Stoechiometric coefficients of 0.37 and 0.32 are also estimated from the Odum curve under low-NO$_x$ and high-NO$_x$ conditions respectively. Moreover, according to the Van Krevelen plot proposed by Chhabra et al. (2011), the most appropriate guaiacol SOA surrogate has an O/C and H/C respectively in the ranges 0.7–1

and 1.2–1.5. With its O/C and H/C ratios of 0.71 and 1.43 ratios, GHDPerox is in the right position of the Van Krevelen plot. The second part of the OH oxidation mechanism for guaiacol follows reactions R19, R20 and R21:

$$RADGUAI + HO_2 \xrightarrow{k_{14}} 0.37\ GHDPerox \tag{R19}$$

$$RADGUAI + NO \xrightarrow{k_{15}} 0.32\ GHDPerox \tag{R20}$$




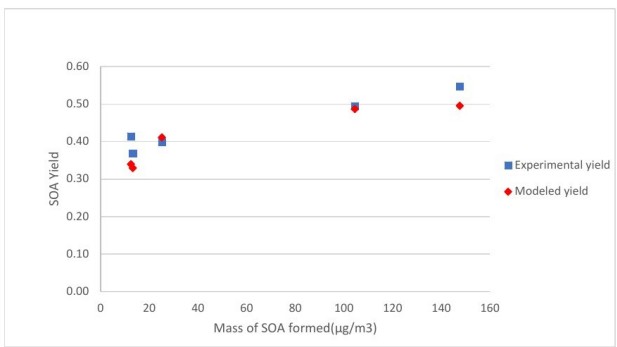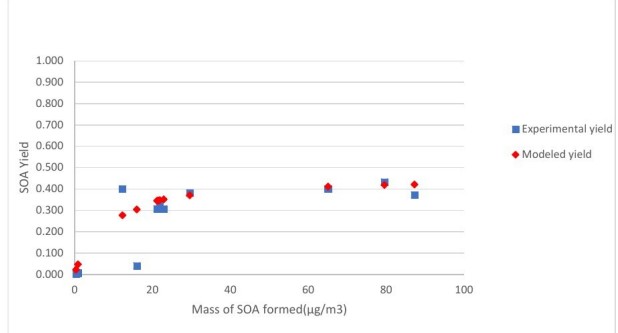

**Figure 4.** SOA experimental and modeled yield data from smog chamber for guaiacol under low-NO$_x$ conditions (left panel) (experimental data from Chhabra et al. (2011); Yee et al. (2013) and under high-NO$_x$ conditions (right panel) (experimental data from Yee et al. (2013); Lauraguais et al. (2012))

$$RADGUAI + NO_3 \xrightarrow{k_{16}} 0.32\ GHDPerox \tag{R21}$$

### 2.1.6 Oxidation of naphthalene and methylnaphthalene

As detailed in Couvidat et al. (2013), data from the chamber experiments of Chan et al. (2009) are used to fit two products
from the oxidation of naphthalene and methylnaphthalene under low-NO$_x$ and high-NO$_x$ conditions. The SOA surrogates
are chosen amongst the compounds detected by Kautzman et al. (2010). Under low-NO$_x$ conditions (reactions with HO$_2$,
the methylperoxy radical MEO$_2$ and the peroxyacetyl radical C$_2$O$_3$), BBPAHlN (C$_6$H$_6$O$_6$, dihydroxyterephthalic acid) is the
surrogate chosen to represent SOA formation from the oxidation of naphthalene and methylnaphthalene. Under high-NO$_x$
conditions, BBPAHhN (C$_8$H$_6$O$_4$, phthalic acid) is the surrogate chosen, because its theoretical saturation vapor pressure (2.04
10$^{-7}$ torr), estimated with SIMPOL.1 (Pankow and Asher, 2008), is the closest to the experimental one (10$^{-6}$ torr) estimated
from the Odum curve plotted by Couvidat et al. (2013). The oxidation reactions leading to SOA formation from naphthalene
and methylnaphthalene are presented in Table B3 of Appendix B.

### 2.1.7 Oxidation of USC>6 compounds

It is not easy to design a chemical mechanism for the structurally assigned and unassigned compounds with at least six carbon
atoms per molecule (USC>6 compounds). In this study, USC>6 compounds are assumed to undergo either the same OH oxidation mechanisms as phenol or as naphthalene, which are previously discussed in sections 2.1.1 and 2.1.6 respectively.

Table B3 in Appendix B summarizes the oxidation reactions added to the chemical mechanism CB05 for each VOC. All
properties of the added compounds are presented in Table B1 of Appendix B. The chemical structure of the SOA compounds



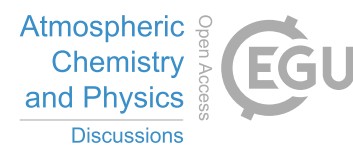

are given in Table B2.

## 2.2 SOA formation from I/S/L-VOCs

Different parameterizations may be used to describe the formation of SOA from the gaseous I/S/L-VOCs emitted from wild-

5 fires, with or without an ageing scheme: one-step oxidation scheme (no ageing) and multi-generational oxidation scheme.

In the one-step oxidation scheme, used for example in Couvidat et al. (2012); Zhu et al. (2016); **?**, the primary surrogates (BBPOAlP, BBPOAmP and BBPOAhP of saturation concentration C*: log(C*)= -0.04, 1.93, 3.5 respectively) undergo one oxidation step in the gas phase, leading to the formation of secondary surrogates (BBSOAlP, BBSOAmP and BBSOAhP). Compared to the primary products, the volatility of the secondary products is reduced by a factor of 100 and their molecular

weight is increased by 40% (Couvidat et al., 2012; Grieshop et al., 2009). Tables in Appendix C list the 3 OH-oxidation reactions and the properties of the primary and secondary surrogates.

For the multi-generational scheme, the VBS approach based on the hybrid VBS (Donahue et al., 2006, 2011; Koo et al., 2014; Giancarlo et al., 2017) is used. In this scheme (Koo et al., 2014; Giancarlo et al., 2017), the basis set uses five volatility surrogates with different saturation concentrations ranging from 0.1 to 1000 $\mu$g.m$^{-3}$. BBPOA0, BBPOA1, BBPOA2, BBPOA3,

BBPOA4 refer to the primary surrogates and BBSOA0, BBSOA1, BBSOA2, BBSOA3 refer to the secondary ones. In the gas phase, the primary and secondary surrogates react with OH at a rate of $4.10^{-11}$ molecule$^{-1}$.cm$^3$.s$^{-1}$ (Donahue et al., 2013). During each oxidation step, the oxidation of the surrogate increases the surrogate oxygen number and decreases its volatility and carbon number, due to functionalization and fragmentation. The reactions and the properties of the surrogates of the multi-generational scheme are shown in Appendix D.

## 20 3 3D simulation over the Mediterranean region

The impact of wildfires on PM concentrations and optical depths in the Euro-Mediterranean during the summer 2007 was studied by Majdi et al. (2018). Here, the CTM Polair3D/Polyphemus (Mallet et al., 2007; Sartelet et al., 2012) is used with a similar set up as Majdi et al. (2018).

Two domains are considered in this study (Figure 5): one nesting domain covering Europe and North Africa and a nested one

over the Mediterranean. The horizontal resolutions used are $0.5° \times 0.5°$ and $0.25° \times 0.25°$ for the nesting and nested domains respectively. The vertical dimension is discretized with 14 levels in Polyphemus (from the ground to 12 km).

The CB05 gas-phase chemical mechanism is used in conjunction with the chemical mechanism H$^2$O to model the formation of SOA from 5 classes of precursors namely: I/S/L-VOCs of anthropogenic emissions, aromatic VOCs, isoprene, monoterpene, sesquiterpenes (Kim et al., 2011; Couvidat et al., 2012). In this work, the SOA mechanism H$^2$O$_{aro}$ developed in section 2.1 for

aromatic VOCs, precursors of SOA, is added. Gas/particle partitioning is modeled using ISORROPIA (Nenes et al., 1999) for inorganics and using SOAP for organics (Couvidat and Sartelet, 2015), assuming thermodynamic equilibrium between gases and particles.



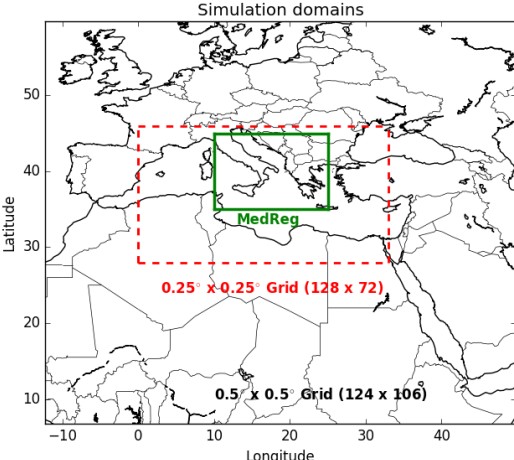

**Figure 5.** Simulation domains including one large domain (with a $0.5° \times 0.5°$ horizontal resolution) and a smaller domain (at a $0.25° \times 0.25°$ horizontal resolution) delimited by the dotted red box. The sub-region MedReg (Balkan + Greece + Eastern Europe + Italy) indicated in green box, is used in this study.

As in Majdi et al. (2018), POA from fire and anthropogenic emissions are assumed to be the condensed phase of I/S/L-VOCs. The gaseous emissions of I/S/L-VOCS from wildfires and their ageing are described in section 2.2.

Dry deposition of gaseous I/S/L-VOCs from wildfires is parameterized based on Wesely (1989), modeling deposition as a serie of resistors consisting of an atmospheric, a laminar sublayer and a bulk surface resistance. The surface resistance is a function of the effective Henry's law constant ($H_{eff}$, M.atm$^{-1}$). For I/S/L-VOCs, this constant varies with the volatility, as detailed in Hodzic et al. (2016). The reactivity factor $f_0$ is set to 0.1 (Karl et al., 2010; Knote et al., 2015). All the parameters used to compute the dry-deposition velocities of the I/S/L-VOCs are summarized in Table E1 of Appendix E.

## 4    Sensitivity simulations

To assess the relative influence of emissions of VOCs, I/S/L-VOCs from wildfires on OA concentrations, six sensitivity simulations are performed. The setup of the different simulations is summarized in Table 1.

The simulation "*onestepISLVOC*" uses the default setup, i.e. the setup used in the previous study (Majdi et al., 2018): for VOC emissions, only toluene and xylene are considered, gaseous I/S/L-VOCs emissions are estimated from POA emissions and their ageing is modeled using a one-step oxidation scheme. The simulation "*MultstepISLVOC*" is conducted to highlight the impact of the ageing scheme of the gaseous I/S/L-VOCs from wildfires on SOA formation. To do so, the multi-generational scheme (Giancarlo et al., 2017) is used for the gaseous I/S/L-VOCs from wildfires.

To assess the impact of VOCs on SOA formation, the Simulation "*Multstep-withVOC*" uses the same setup as the simulation "*MultstepISLVOC*" but all the VOCs, which are SOA precursors are added to the model, as detailed in section 5. Because the





relative impact of I/S/L-VOCs on OA formation depends on how gaseous I/S/L-VOCs emissions are computed, the simulation "*Multstep-UnNMOG-withVOC*" is the same as the simulation "*Multstep-withVOC*" but the gaseous I/S/L-VOC emissions are calculated from NMOG.

The sensitivity of two parameters involved in the modeling of the ageing of these VOCs is also assessed: the enthalpy of
5   vaporization ($\Delta H_{vap}$) of the SOA formed from the oxidation of the VOCs and the SOA formation mechanism from USC>6 compounds.

Several studies consider $\Delta H_{vap}$ of the formed SOA as constant (Sheehan and Bowman 2001; Donahue et al. 2005; Stanier et al. 2007). For SOA from $\alpha$-pinene, Donahue et al. (2005) estimated $\Delta H_{vap}$ to be about 30 kJ.mol$^{-1}$. This is lower than the $\Delta H_{vap}$ values calculated for individual components using SIMPOL.1. Stanier et al. (2007) also estimated $\Delta H_{vap}$ to be
10   in the range of 10–50 kJ.mol$^{-1}$. In the simulation "*Multstep-withVOC-Enthalpy-SIMPOL.1*", the enthalpy of vaporization is calculated for the SOA surrogates formed from VOCs using SIMPOL.1 rather than being constant as in the simulation "*Multstep-withVOC*". In the simulation "*Multstep-withVOC-USC>6naph*", the SOA formation mechanism from USC>6 compounds is taken as the formation mechanism of naphtalene, rather than being the same as the formation mechanism of phenol in the simulation "*Multstep-withVOC*".

**Table 1.** Summary of the sensitivity simulations performed by Polyphemus. (N/A: not applicable)

| Simulations | **Wildfires** | | | | |
| --- | --- | --- | --- | --- | --- |
| | **gaseous VOCs emissions** | **I/S/L-gaseous I/S/L-VOCs ageing** | **added VOCs precursors** | **$\Delta H_{vap}$ (kJ.mol$^{-1}$)** | **USC>6 mechanism** |
| *OnestepISLVOC* | from POA | one-step | no | N/A | N/A |
| *MultstepISLVOC* | from POA | multi-generational | no | N/A | N/A |
| *Multstep-withVOC* | from POA | multi-generational | yes | 50 | phenol mechanism |
| *Multstep-UnNMOG-withVOC* | from NMOG | multi-generational | yes | 50 | phenol mechanism |
| *Multstep-withVOC-Enthalpy-SIMPOL.1* | from POA | multi-generational | yes | SIMPOL.1 | phenol mechanism |
| *Multstep-withVOC-USC>6naph* | from POA | multi-generational | yes | 50 | naphthalene mechanism |





## 5   Emissions of SOA precursors from wildfires

To better understand the contribution of $OA_{tot}$ precursors emitted by wildfires and their relative importance for $OA_{tot}$ and OA formation, the estimation of $OA_{tot}$ precursors emissions is first detailed. Two categories of SOA precursors are distinguished depending on their volatilities: VOCs and gaseous I/S/L-VOCs.

### 5.1   VOC emissions

Bruns et al. (2016) identified the most significant gaseous VOC precursors of SOA from residential wood combustion and presented their contribution to SOA concentrations. In this work, VOC precursors emitted from wildfires are chosen based on the list of Bruns et al. (2016), their emission factors for wildfires and SOA yields. Toluene, xylene, phenol, benzene, catechol, cresol, furan, naphthalene, methylnaphthalene and the structurally assigned and unassigned compounds with at least 6 carbon atoms per molecule (USC>6 compounds) are retained. Table A1 in Appendix A shows the VOCs, the corresponding SOA yields and emission factors from fires of various vegetation types.

Daily fire emissions of toluene, xylene, phenol, benzene and furan are estimated by the APIFLAME fire emission model (Turquety et al., 2014). The emission of factors in Akagi et al. (2011) are used to calculate the emissions of each species from the carbon emissions. The emission factors of toluene, xylene, benzene, furan and phenol are available in the Akagi et al. (2011) inventory and provided in term of g species per kg dry biomass burned for different standard vegetation types (temperate forest, crop residues, pasture maintenance, savanna and chaparral). Using an aggregation matrix, emissions of these inventory VOCs are converted to model species.

However, cresol, catechol, syringol, guaiacol, naphthalene, methylnaphthalene emission factors are missing from Akagi et al. (2011) inventory. For cresol, catechol, guaiacol and syringol, these emission factors are calculated from the molar emission ratio to phenol, and for naphthalene and methylnaphthalene, they are calculated from the molar emission ratio to benzene (Stockwell et al., 2015) following equation (2):

$$EF_i = ER_{mass,i}.EF_x = \left( ER_{mol,i}.\frac{M_{w,i}}{M_{wx}} \right).EF_x \tag{2}$$

where $i$ represents a VOC (cresol, catechol, guaiacol, syringol, naphthalene and methylnaphthalene), $ER_{mass,i}$ is the mass emission ratio of the VOC $i$ to phenol or benzene, $EF_x$ is the mass emission factor of phenol or benzene (determined using APIFLAME), $ER_{mol,i}$ is the molar emission ratio of the VOC $i$ (cresol, catechol, guaiacol, syringol, naphthalene and methyl-naphthalene), $M_{w,i}$ is the molar weight of the VOC $i$, $M_{wx}$ is the molar weight of phenol (= 90 g.mol$^{-1}$) or benzene (= 78 g.mol$^{-1}$).

For two types of vegetation $j$ (chaparral and crop residue), the emission ratios $ER_{mol,i,j}$ are obtained from Stockwell et al. (2015). Then in each model grid cell, the emission ratio of the VOC $i$ (cresol, catechol, guaiacol, syringol, naphthalene or methylnaphthalene) to phenol or benzene is obtained by weighting the emission ratios over the burned vegetation types:

$$ER_{mol,i} = \sum_{j=1}^{n} Fveg_j.ER_{mol,i,j} \tag{3}$$



where Fveg$_j$ is the burning fraction for each vegetation type, $ER_{mol,i,j}$ is the emission ratio of the VOC $i$ to phenol or benzene for each vegetation type.

Considering only these two types of vegetation (crop residue and chaparral), for which emission ratios are available, may lead to an underestimation of the emissions factors and therefore the emissions of cresol, catechol, guaiacol, syringol, naphthalene and methylnaphthalene emissions. Indeed, Figure 6 shows the percentages of the different vegetation types in the burned area detected over the sub-region MedReg. Chaparral and crop residue make only 29.5% of burned areas detections. Savanna and temperate forest are considered as the dominant vegetation types detected in the burned areas and their contributions to burned area detections reach 32.7% and 37.2% respectively. Therefore, neglecting the emission factors for temperate forest and savanna would lead to a significant underestimation of the SOA precursor emissions. Because the $EF$ of VOCs emitted by wildfires of crop residue, chaparral, temperate forest and savanna in the inventory of Akagi et al. (2011) are often of the same order of magnitude (Table A1 of Appendix A), it is assumed here that temperate forest and savanna have the same EF as chaparral for cresol, catechol, guaiacol, syringol, naphthalene and methylnaphthalene.

According to Bruns et al. (2016), the structurally assigned and unassigned compounds with at least six carbon atoms per molecule (USC>6 compounds) are expected to contribute to SOA formation based on their structures but their SOA yields are unknown. In this work, USC>6 compounds emissions are deduced by multiplying phenol emissions by a factor of 1.7, deduced from the ratio of the SOA contribution of USC>6 compounds to the SOA contribution of phenol (Bruns et al., 2016).

## 5.2  I/S/L-VOC emissions

The gaseous I/S/L-VOC emissions from wildfires are estimated either from the POA emissions released from wildfires, by multiplying them by a constant ratio of I/S/L-VOC/POA=1.5 (Kim et al., 2016), or from the unspeciated NMOG released from wildfires (Jathar et al., 2014). The fraction of unspeciated NMOG is estimated as the difference between the total NMOG emissions from Akagi et al. (2011) inventory and the VOC emissions, which represent the sum of the total identified NMOG in the Akagi et al. (2011) inventory, plus the VOCs previously added to the Akagi et al. (2011) inventory (cresol, catechol, guaiacol, syringol, naphthalene, methylnaphthalene and USC>6 compounds). In this work, as in Jathar et al. (2017), these unspeciated NMOG are assumed to be gaseous I/S/L-VOCs. They represent 36% of the total NMOG emissions, which is consistent with the work of Yokelson et al. (2013), which estimates that between 35% to 64% of NMOG are the gaseous I/S/L-VOCs. Similarly to anthropogenic emissions (detailed in section 3), the gaseous I/S/L-VOC emissions from wildfires are distributed into three volatility bins depending on their saturation concentration (C$^*$): low volatility (BBPOAlP, log(C$^*$)= -0.04), medium volatility (BBPOAmP, log(C$^*$)= 1.93) and high volatility (BBPOAhP, log(C$^*$)= 3.5). The volatility distribution at emission is 25%, 32%, and 43% for BBPOAlP, BBPOAmP and BBPOAhP respectively (Couvidat et al., 2012; May et al., 2013; Giancarlo et al., 2017).

## 5.3  Emissions over the Mediterranean domain

The left panel of Figure 7 presents the emissions of total (gas+particle) OA$_{tot}$ precursors (VOCs, I/S/L-VOCs) for the different sensitivity simulations, spatially and temporally averaged over the sub-region MedReg (Figure 5) and during the summer





Vegetation types in the burned area detected in MedReg (%)

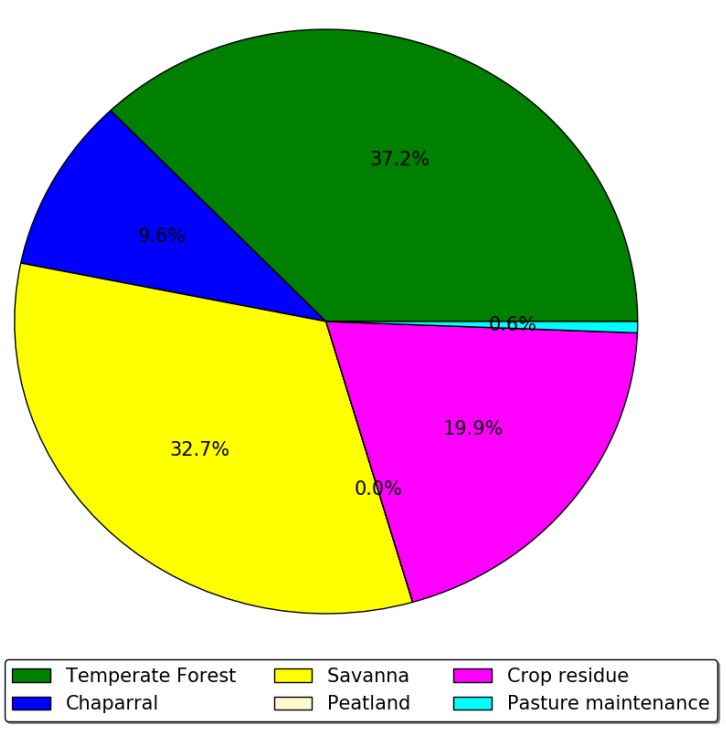

**Figure 6.** Percentage of the different vegetation types in the burned area detected over the sub-region MedReg during the summer 2007.

2007. The emissions of VOCs and I/S/L-VOCs are similar in all the sensitivity simulations except for the simulation *Multstep-UnNMOG-withVOC* which estimates the gaseous I/S/L-VOCs emissions from NMOG. The emissions of gaseous I/S/L-VOCs estimated from NMOG emissions are slightly lower than those estimated from POA emissions. The emissions of gaseous I/S/L-VOCs (estimated from POA or from NMOG) are higher by a factor of about 2.5 than the emissions of VOCs.

5    The spatial distribution of the relative contribution of VOCs to gaseous precursors emissions (I/S/L-VOCs from NMOG + VOCs) is assessed in Figure 8. Emissions of wildfires occur mostly over Balkans, Greece, Southern Italy, Eastern Europe and Northern Algeria, with a relative contribution of VOCs mostly between 20% and 40%. Locally, over Balkans, the contribution of VOCs can be higher (between 40% and 60%). Figure 9 shows the number of burned area detections for temperate forest. The high contribution of VOCs in Balkans is probably explained by the high number of burned areas detected for temperate

10   forest, which is considered one of the dominant vegetation type in the burned areas.




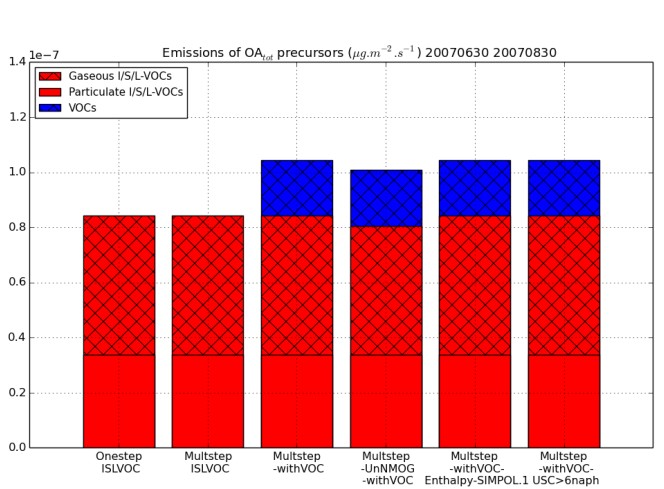
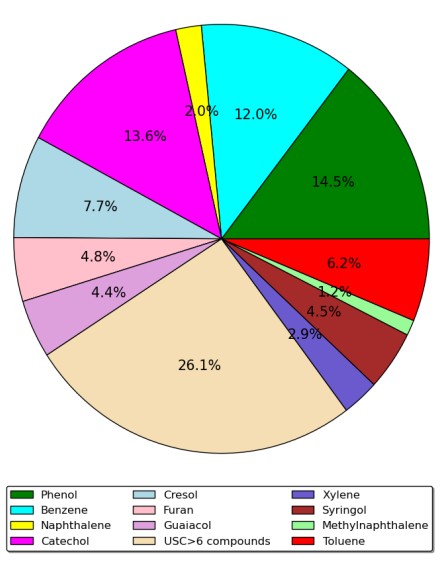

**Figure 7.** Emissions of the $OA_{tot}$ precursors from wildfires for the different sensitivity simulations (left panel) and percentage of emissions for each VOC (right panel) over the sub-region MedReg during the summer 2007.

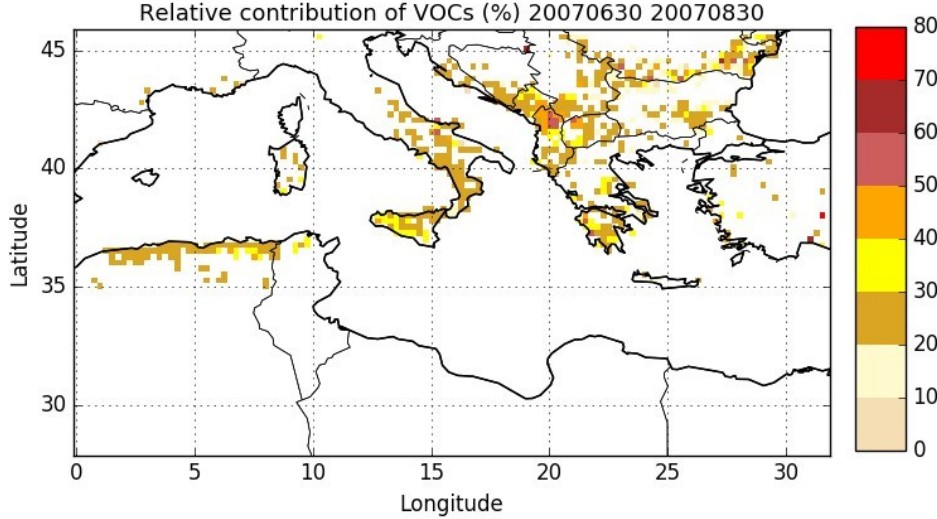

**Figure 8.** Relative contribution of VOCs to gaseous precursors (VOCs + gaseous I/S/L-VOCs) (%) emitted by wildfires over the Mediterranean area during the summer 2007.




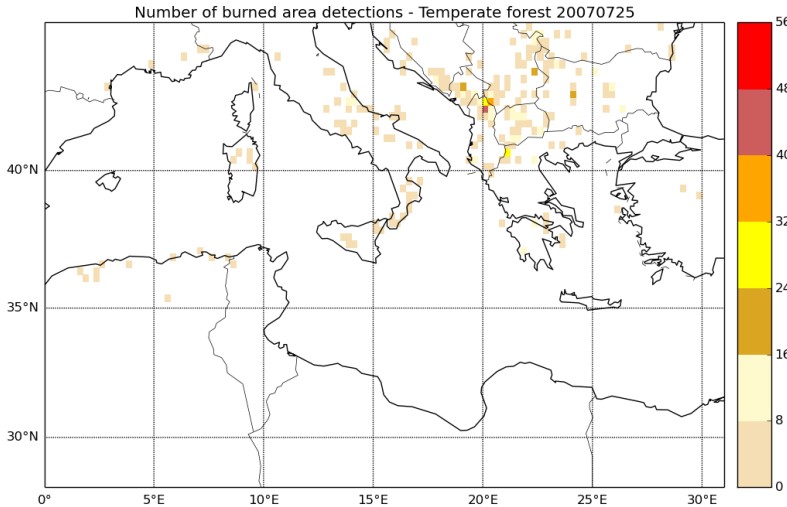

**Figure 9.** Number of burned area detections for temperate forest on 25 July 2007.

The right panel of Figure 7 shows the distribution of VOCs between the different compounds emitted over the sub-region MedReg during the summer 2007. USC>6 compounds dominate (26.1%) followed by phenol (14.5%), catechol (13.6%), benzene (12%), toluene (7%), furan (5%) and cresol (4%). The other VOCs (SOA precursors) contribute to 3% or less of the VOC emissions.





## 6 Results and discussion

The influence of VOCs, I/S/L-VOCs on OA and $OA_{tot}$ concentrations is discussed in this section, as well as the sensitivity to some parameters for OA and $OA_{tot}$ formation from VOCs and gaseous I/S/L-VOCs.

### 6.1 Influence on OA concentrations

Figure 10 presents the $OA_{tot}$ concentrations from different precursors emitted by biomass burning (VOCs, I/S/L-VOCs). The contributions of the different $OA_{tot}$ precursors from different simulations are compared. In the simulation *Multstep-withVOCs*, the precursors are VOCs, I/S/L-VOCs with gaseous emissions estimated from POA) and with ageing by the multi-step oxidation scheme. In the simulations *onestep-ISLVOCs* and *Multstep-ISLVOC*, the precursors are I/S/L-VOCs with gaseous emissions estimated from POA emissions and with ageing by the one-step and the multi-step oxidation schemes respectively. In the simulation *Multstep-unNMOG-withVOCs*, the precursors are VOCs and I/S/L-VOCs with gaseous emissions estimated from NMOG emissions and with ageing by the multi-step oxidation scheme.

The emissions of VOCs, are lower than those of gaseous I/S/L-VOCs estimated from NMOG (or POA) emissions by almost a factor of about 2.5. This preponderance of I/S/L-VOCs is observed not only for emissions but also for concentrations. The primary and secondary OA concentrations from gaseous I/S/L-VOCs (estimated from NMOG emissions and from POA emissions) are about 10 times higher than the OA concentrations from VOCs. Most of the OA and $OA_{tot}$ concentrations are formed from I/S/L-VOCs (about 90% and 75% respectively). The OA concentrations are slightly higher (by about 10%) when the gaseous I/S/L-VOCs are estimated from POA rather than from NMOG emissions. This difference corresponds to the difference observed in emissions (gaseous I/S/L-VOC emissions estimated from POA are slightly higher than those estimated from NMOG).

28 to 42% of the OA concentrations from I/S/L-VOCs emissions are primary. The amount of POA from I/S/L-VOCs emissions in simulation *onestep-ISLVOCs* (28%) is lower than the one in the simulation *Multstep-ISLVOC* (42%) because of the differences in the volatility properties of the species in the two ageing schemes.

The OA concentrations simulated with the one-step and the multi-generational schemes are nearly similar (about 5% difference). However, the primary and secondary $OA_{vapor}$ concentrations (the gas-phase of $OA_{tot}$ concentrations) are lower with the multi-generational scheme because of fragmentation.

A large part of $OA_{tot}$ concentrations from VOCs (∼70%) is in the gas phase. This suggests that the influence of the VOC emissions on particle OA concentrations could be larger if the surrogates from these VOC oxidations partition more easily to the particle phase. This could be the case if further ageing mechanisms are considered for these VOCs or if the particles are very viscous (Kim et al., 2018).

Using the SOA formation mechanism of naphthalene rather than the SOA formation mechanism of phenol affects slightly the $OA_{tot}$ concentrations from VOCs (∼ 3%). Similar results are found when calculating the enthalpy of vaporization of the





formed SOA with SIMPOL.1 instead of using a constant ($\Delta H$=50 kJ.mol$^{-1}$). This shows that the SOA formation from VOCs is poorly sensitive to these parameters involved in the modeling of the VOCs ageing.

Figure 11 presents the contribution of VOCs to biomass-burning OA concentrations, as simulated by the simulation *Multstep-withVOCs*. In agreement with the preponderance of the contribution of I/S/L-VOCs discussed above, the VOC contribution is

5 between 10% and 25% in most of the Mediterranean where biomass-burning OA concentrations are above 1 $\mu$g m$^{-3}$. A larger contribution of VOCs (reaching 30%) is observed in the Balkans, where the biomass-burning OA concentrations are the highest, with a large fraction of temperate forests burning.

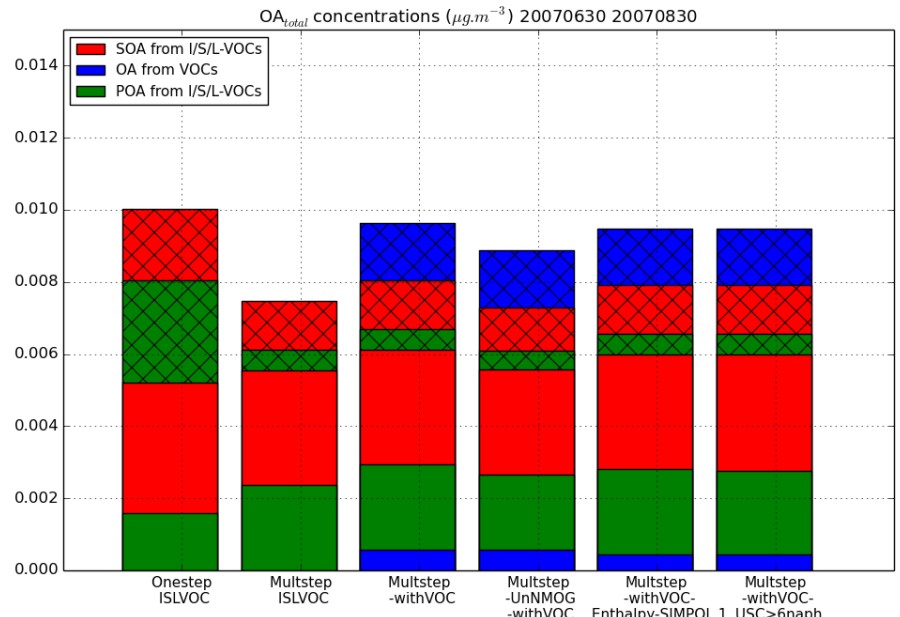

**Figure 10.** Mean surface OA$_{tot}$ concentrations from different OA$_{tot}$ precursors over the subregion MedReg for each sensitivity simulation. The cross-hatched part corresponds to OA concentrations in the gaseous phase, while the plain parts correspond to OA concentrations in the particle phase.

Figure 12 shows the distribution of the OA concentrations formed from the different VOCs emitted by wildfires over the

10 sub-region MedReg during the summer 2007. The largest contribution comes from phenol, benzene and catechol. It represents about 47% of the OA concentrations from VOCs, and 40% of the VOC emissions. The second largest contribution comes from USC>6 compounds. It represents about 23% of the OA concentrations from VOCs, and 26% of the VOC emissions. Toluene and xylene, which were taken into account in the previous version of the model, have a high yield compared to other VOCs. They make about 12% of the OA concentrations from VOCs, whereas their emissions represent about 9% of the VOC

emissions. Furan, which makes about 5% of VOC emissions, does not contribute to OA concentrations (contribution lower than





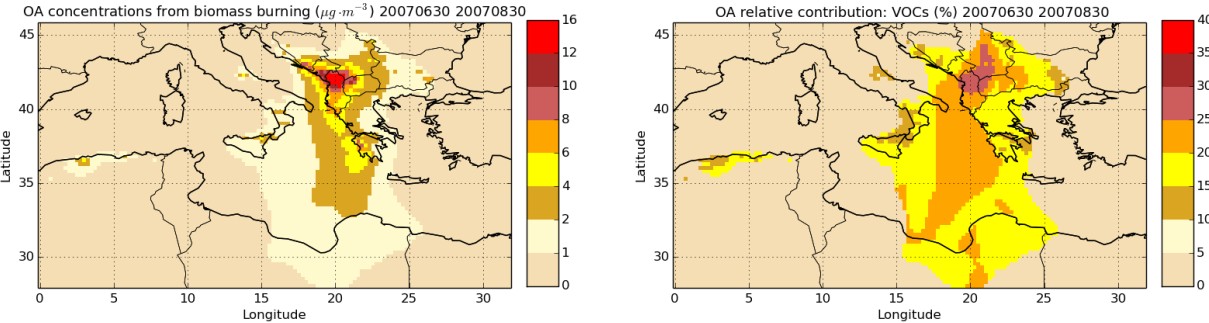

**Figure 11.** Daily mean surface OA concentrations from wildfires (left panel) and the relative contribution of VOCs (%) to OA from wildfires (right panel) during the summer 2007.

1%). Cresol contributes equally to VOC emissions and SOA concentrations (about 7%). Syringol, which contributes to only 4% of VOC emissions, contribute to about 6% of the OA concentrations. The other VOCs (naphthalene, methylnaphthalene, guaiacol) have a low contribution (equal to or lower than 3%).

### 6.1.1 Sensitivity of PM$_{2.5}$ concentrations

To assess the sensitivity of PM$_{2.5}$ concentrations to VOCs and gaseous I/S/L-VOCs and parameters related to their emissions or ageing, differences of PM$_{2.5}$ concentrations among the sensitivity simulations are compared. The sensitivity to the gaseous I/S/L-VOC ageing scheme is assessed by computing relative differences between the simulations *OnestepISLVOC* and *MultstepISLVOC*). The sensitivity to the gaseous I/S/L-VOCs emissions is assessed by computing relative difference between the simulations *Multstep-withVOCs* and *Multstep-unNMOG-withVOC*). The sensitivity to the VOC emissions is assessed by
computing the relative difference between the simulations *Multstep-withVOCs* and *MultstepISLVOC*.

     Figure 13 shows the average PM$_{2.5}$ concentrations, as well as relative differences of PM$_{2.5}$ concentrations among the sensitivity simulations. The PM$_{2.5}$ concentrations are especially high with average concentrations above 20 $\mu$g m$^{-3}$ where wildfires occur especially in the Balkans and Greece. Majdi et al. (2018) studied the simulation *OnestepISLVOC* and found that comparing to PM$_{2.5}$ observations, the model tends to underestimate PM$_{2.5}$ concentrations (MFB=-32%). Moreover, they highlighted
that surface PM$_{2.5}$ concentrations are sensitive to gaseous I/S/L-VOCs emissions and their impact on surface PM$_{2.5}$ concentrations over the fire regions can reach 10-20% in the fire plume and 30% locally.

     Concerning the influence of the gaseous I/S/L-VOCs ageing scheme, the relative differences between the simulations *OnestepISLVOC* and *MultstepISLVOC* are low (below 5%). The differences can be positive or negative, because the one-step oxidation scheme and the multi-step oxidation schemes lead to SOA of different volatilities. The sign of the differences de-




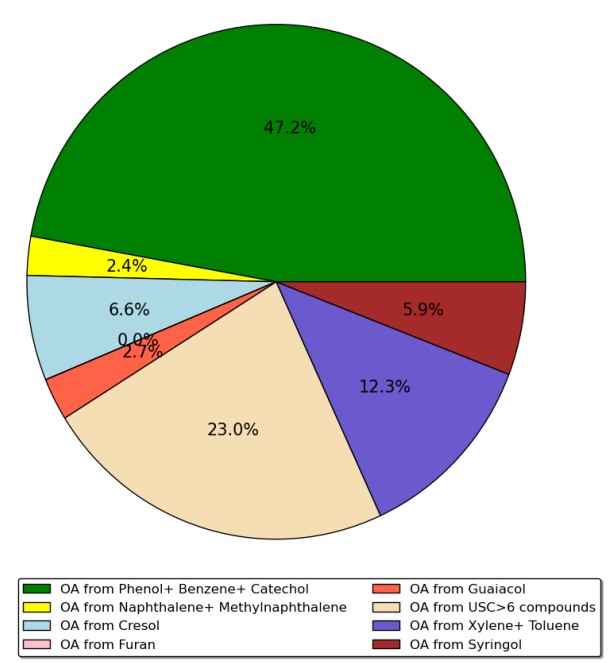

**Figure 12.** Distribution of the OA concentrations formed from the different VOCs emitted by wildfires over the sub-region MedReg during the summer 2007.

pends on the SOA volatilities and on the partitioning between the gas and the particle phases of I/S/L-VOCS, which itself depends on $PM_{2.5}$ concentrations. The comparison of the relative difference of $PM_{2.5}$ concentrations between the simulations *OnestepISLVOC* and *MultstepISLVOC* (upper left panel) and the daily mean $PM_{2.5}$ concentrations (lower right panel) shows that the differences tend to be positive (higher concentrations with multi-generational ageing than with one-step ageing) in the regions of strong fires where $PM_{2.5}$ concentrations are high, and negative in the fire plume where $PM_{2.5}$ concentrations are lower.

The emissions of the added VOCs (namely benzene, phenol, cresol, catechol, furan, guaiacol, syringol, naphthalene, methyl-naphthalene, the structurally assigned and unassigned compounds with at least 6 carbon atoms per molecule (USC>6) lead to a moderate increase of $PM_{2.5}$ concentrations (up to 25% in the Balkans) (lower left panel). $PM_{2.5}$ concentrations are more sensitive to the parameterization used to estimate the gaseous I/S/L-VOC emissions. Estimating the gaseous I/S/L-VOCs emissions from POA rather than from NMOG and results in higher local $PM_{2.5}$ concentrations (+8 to +16% in Greece) and lower $PM_{2.5}$ concentrations mainly in Balkans (-30%) and in the fire plume (-8 to -16%). The larger fraction of $PM_{2.5}$ concentrations is shown in Balkans where the gaseous I/S/L-VOCs emissions from NMOG are higher than those emitted from POA. This





is explained by differences in NMOG and POA emissions. Figure 14 shows daily mean emissions of POA and NMOG from wildfires during summer 2007. The main difference between POA and NMOG emissions are located in Balkans, where the largest fraction of burned temperate forest is observed. In Akagi et al. (2011), the emission factor of POA is unavailable for temperate forest. This may be explained by the lower POA emissions in Balkans.

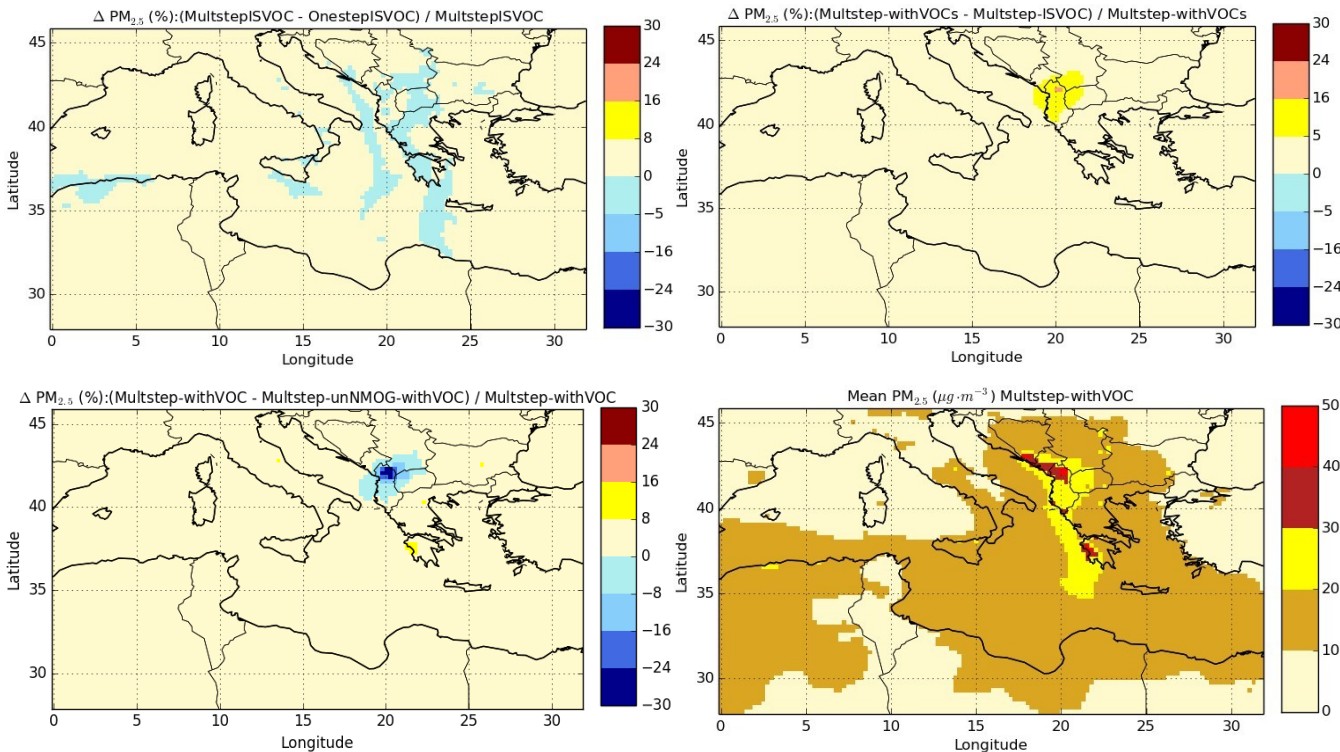

**Figure 13.** Sensitivity of surface PM$_{2.5}$ concentrations to the gaseous I/S/L-VOCs ageing scheme (upper left panel), the SOA from the selected VOC (upper right panel), the SOA from gaseous I/S/L-VOCs emissions estimated from NMOG (lower left panel) and daily mean PM$_{2.5}$ concentrations from the $Multstep - withVOC$ simulation (lower right panel) during the summer 2007 (from 30 June to 30 August 2007).



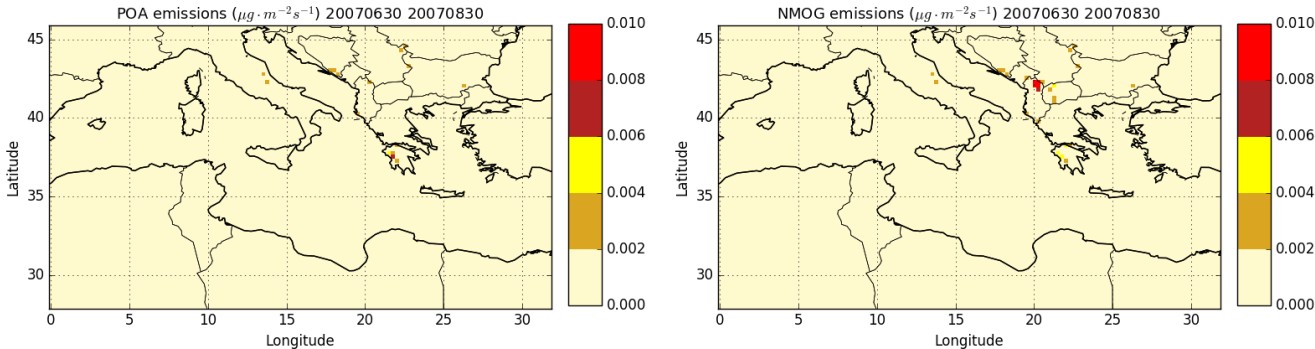

**Figure 14.** Daily mean POA (left panel) and NMOG (right panel) emissions from wildfires during the summer 2007.

## 7 Conclusion

This study quantified the relative contribution of $OA_{tot}$ precursors (VOCs, I/S/L-VOCs) emitted by wildfires to OA formation and particle concentrations, during the summer 2007 over the Euro-Mediterranean region. A new chemical mechanism $H^2O_{aro}$ was developed to represent the SOA formation from selected VOCs, namely toluene, xylene, benzene, phenol, cresol, catechol, furan, guaiacol, syringol, naphthalene, methylnaphthalene, the structurally assigned and unassigned compounds with at least 6 carbon atoms per molecule (USC>6), based on smog chamber experiment under low and high-$NO_x$. This mechanism was implemented in the chemistry transport model Polair3D of the air-quality platform Polyphemus. The gaseous I/S/L-VOCs emissions were estimated either from POA emissions using a factor of 1.5 or from NMOG using a factor of 0.36. Over the Euro-Mediterranean area, the OA concentrations emitted by wildfires originate mostly from I/S/L-VOCs.

The OA concentrations from gaseous I/S/L-VOCs are about 10 times higher than the OA concentrations from VOCs. However, a large part of OA concentrations from VOCs is in the gas phase ($\sim 70\%$), and the contribution of the oxidation of VOCs to the OA concentrations is locally significant (can reach 30% in Balkans). This suggests that the influence of the VOC emissions on OA concentrations could be large, and as significant as the influence of gaseous I/S/L-VOC emissions, if the surrogates from these VOC oxidations partition more easily to the particle phase. This could be the case if further ageing mechanisms are considered for these VOCs or if the particles are very viscous (Kim et al., 2018).





# Appendix A

**Table A1.** The VOC that are SOA precursors and their emission factors ($EF$) and SOA yields. [a]Yee et al. (2013). [b]Nakao et al. (2011). [c]Ng et al. (2007). [d]Gómez et al. (2008). [e]Chan et al. (2009). [f]Chhabra et al. (2011). [g]Pereira et al. (2009). [1]Akagi et al. (2011). [1] Emission ratio ($ER$) of the VOC to phenol from Stockwell et al. (2015). [2] Emission ratio of the VOC to benzene from Stockwell et al. (2015). EF from Akagi et al. (2011) are in black, ER from Stockwell et al. (2015) are in blue and EF in red are deduced from the assumption considering that temperate forest and savanna have the same EF as chapparal.

| VOCs | EF*(g/kg) | | | | | $Y_{SOA}$ | $NO_x$ regime |
|---|---|---|---|---|---|---|---|
| | Savanna | Crop residue | Pasture Maintenance | Temperate forest | Chaparral | | |
| Phenol | 0.52 | 0.52 | 1.68 | 0.33 | 0.45 | $0.44^a$ | low/high $NO_x$ |
| Cresol | $0.26^1$ | $0.35^1$ | - | $0.26^1$ | $0.26^1$ | $0.36^b$ | low $NO_x$ |
| Benzene | 0.20 | 0.15 | 0.70 | - | - | $0.33^c$ | low/high $NO_x$ |
| Catechol | $0.90^1$ | $0.48^1$ | - | $0.90^1$ | $0.90^1$ | $0.39^b$ | low $NO_x$ |
| Furan | 0.17 | 0.11 | 2.63 | 0.2 | 0.18 | $0.05^e$ | high $NO_x$ |
| Syringol | $0.27^1$ | $0.23^1$ | - | $0.27^1$ | $0.27^1$ | $0.26^{a,f}$ | medium-high $NO_x$ |
| Guaiacol | $0.27^1$ | $0.81^1$ | - | $0.27^1$ | $0.27^1$ | $0.45^{a,f}$ | medium $NO_x$ |
| Naphthalene | $0.16^2$ | $0.31^2$ | - | $0.16^2$ | $0.16^2$ | $0.52^{e,f}$ | medium $NO_x$ |
| Methylnaphthalene | $0.06^2$ | $0.22^2$ | - | $0.06^2$ | $0.06^2$ | $0.52^{e,f}$ | medium-low $NO_x$ |
| Toluene | 0.08 | 0.19 | 0.34 | - | - | $0.24^{c,g}$ | low/high $NO_x$ |
| Xylene | 0.01 | - | - | 0.11 | - | $0.20^{c,f}$ | low/high $NO_x$ |





## Appendix B

**Table B1.** Properties of the compounds added to the model.

| Species | Species names | Molecular formula | Mw$^a$ | $\Delta H_{vap}$$^b$ | $P_{sat}$$^c$ | $K_p$$^d$ | H$^e$ |
|---|---|---|---|---|---|---|---|
| PHEN | phenol | $C_6H_6O$ | 94 | 60.88 | 99.99 $10^2$ | 1.98 $10^{-6}$ | - |
| CAT | catechol | $C_6H_6O_2$ | 110 | 76.91 | 6.5 $10^{-4}$ | 2.57 $10^{-4}$ | - |
| ACIDMAL | maleylacetic acid | $C_6H_6O_5$ | 158 | 81.66 | 4.59 $10^{-8}$ | 2.56 | 8.68 $10^{11}$ |
| BENZ | benzene | $C_6H_6$ | 78 | 43.25 | 15.23 | 1.30 $10^{-8}$ | - |
| CRESp | cresol | $C_7H_8O$ | 108 | 64.53 | 3.98 $10^{-6}$ | 3.75 $10^{-12}$ | - |
| MCAT | methylcatechol | $C_7H_8O_2$ | 124 | 81.36 | 2.46 $10^{-4}$ | 6.08 $10^{-4}$ | - |
| DHMB | dihydroxymethylbenzoquinone | $C_7H_6O_4$ | 154 | 81.73 | 3.52 $10^{-6}$ | 3.4 $10^{-2}$ | 3.62 $10^9$ |
| FUR | furan | $C_4H_4O$ | 68 | 27.45 | 5.925 $10^2$ | 2.5 $10^{-7}$ | - |
| ButDial | butendial | $C_4H_4O_2$ | 84 | 54.03 | 1.89 | 1.17 $10^{-7}$ | - |
| RADButenalCOO | radical | $C_4H_3O_3$ | 99 | - | - | - | - |
| ButenalCOOH | malealdehydic acid | $C_4H_4O_3$ | 100 | 66.92 | 0.0122 | 1.53 $10^{-5}$ | - |
| RADButenCOOHCOO | radical | $C_4H_3O_4$ | 115 | - | - | - | - |
| Buten2COOH | maleic acid | $C_4H_4O_4$ | 116 | 79.83 | 7.803 $10^{-5}$ | 0.00238 | 1.03 $10^9$ |
| SYR | syringol | $C_8H_{10}O_3$ | 154 | 77.41 | 5.49 $10^{-4}$ | 0.0002195 | - |
| GUAI | guaiacol | $C_7H_8O_2$ | 124 | 68.89 | 7.41 $10^{-3}$ | 2.02 $10^{-3}$ | - |
| RADSYR | radical | $C_8H_9O_3$* | 171 | - | - | - | - |
| RADGUAI | radical | $C_7H_7O_2$* | 141 | - | - | - | - |
| PSYR | syringol SOA | $C_8H_{10}O_5$ | 186 | 96.25 | 7.53 $10^{-6}$ | 1.294 $10^{-2}$ | 1.45 $10^{+9}$ |
| GHDPerox | guaiacol SOA (hydroperoxide) | $C_7H_{10}O_5$ | 174 | 99.52 | 5.41 $10^{-7}$ | 0.1972 | 9.89 $10^{+9}$ |
| NAPH | naphthalene | $C_{10}H_8$ | 128 | 61.38 | 0.0398 | 3.64 $10^{-6}$ | - |
| NAPHP | radical | $C_{10}H_7$* | 127 | - | - | - | - |
| MNAPH | methylnaphthalene | $C_{11}H_{10}$ | 142 | 65.26 | 0.0150 | 8.73 $10^{-6}$ | - |
| MNAPHP | radical | $C_{11}H_9$* | 141 | - | - | - | - |
| BBPAHlN | dihydroxyterephthalic acid | $C_8H_6O_6$ | 198 | 131.62 | 1 $10^{-12}$ | 93817.62.59 | 1.65 $10^{+19}$ |
| BBPAHhN | phthalic acid | $C_8H_6O_4$ | 166 | 97.95 | $10^{-6}$ | 97.95 | 1.49 $10^{+9}$ |



| | | | | | | | |
|---|---|---|---|---|---|---|---|
| USC>6$_{phen}$ | - | - | 94 | 60.88 | 99.99 10$^2$ | 1.98 10$^{-6}$ | - |
| USC>6CAT | catechol | $C_6H_6O_2$ | 110 | 76.91 | 6.5 10$^{-4}$ | 2.57 10$^{-4}$ | - |
| USC>6ACIDMAL | maleylacetic acid | $C_6H_6O_5$ | 158 | 81.66 | 4.59 10$^{-8}$ | 2.56 | 8.68 10$^{11}$ |
| USC>6$_{naph}$ | - | $C_{10}H_8$ | 128 | 61.38 | 0.0398 | 3.64 10$^{-6}$ | - |
| USC>6NAPHP | radical | $C_{10}H_7{}^*$ | 127 | - | - | - | - |
| USC>6BBPAHlN | dihroxyterephthalic acid | $C_8H_6O_6$ | 198 | 131.62 | 10$^{-12}$ | 93817.62.59 | 1.65 10$^{+19}$ |
| USC>6BBPAHhN | phthalic acid | $C_8H_6O_4$ | 166 | 97.95 | 10$^{-6}$ | 50 | 1.49 10$^{+9}$ |

$^a$Molar weight (g.mol$^{-1}$)

$^b$Enthalpy pf vaporization (kJ.mol$^{-1}$)

$^c$Saturation vapor pressure (torr)

5  $^d$Partitioning constant (m$^3$.g$^{-1}$)

$^e$Henry's law constant (M.atm$^{-1}$)



**Table B2.** Chemical structure of SOA compounds considered in this study.

| SOA species | chemical structure |
| --- | --- |
| ACIDMAL | |
| DHMB | |
| Buten2COOH | |
| PSYR | |
| GHDPerox | |
| BBPAHlN | |
| BBPAHhN | |



**Table B3.** Reactions leading to SOA formation added to CB05.

| Reactions | Kinetic Rate Parameter ($molecule^{-1}.cm^3.s^{-1}$) |
|---|---|
| **PHEN + OH → 0.75 CAT + OH** | $4.7\ 10^{-13}$ exp(1220/T) |
| **CAT + OH → 0.28 ACIDMAL + OH** | $9.9\ 10^{-10}$ |
| **BENZ + OH → 0.53 PHEN + OH** | $2.3\ 10^{-12}$ exp(-190/T) |
| **CRESp + OH → 0.73 MCAT+ OH** | $4.65\ 10^{-10}$ |
| **MCAT + OH → 0.39 DHMB + OH** | $2\ 10^{-10}$ |
| **FUR + OH → 0.87 ButDial + OH** | $4.19\ 10^{-11}$ |
| **ButDial + OH → 0.83 RADButenalCOO + OH** | $5.20\ 10^{-11}$ |
| **RADButenalCOO + HO2 → 0.15 ButenalCOOH + HO2** | $5.20\ 10^{-13}$ exp(980/T) |
| **RADButenalCOO + NO → NO** | $7.5\ 10^{-12}$ exp(290/T) |
| **RADButenalCOO + XO2 → 0.3 ButenalCOOH + XO2** | $1.0\ 10^{-11}$ |
| **ButenalCOOH + OH → 0.3 RADButenCOOHCOO + OH** | $2.12\ 10^{-11}$ |
| **RADButenCOOHCOO + HO2 → 0.15 Buten2COOH + HO2** | $5.20\ 10^{-13}$ exp(980/T) |
| **RADButenCOOHCOO + NO → NO** | $7.50\ 10^{-12}$ exp(980/T) |
| **RADButenCOOHCOO + XO2 → 0.3 Buten2COOH + XO2** | $1.0\ 10^{-11}$ |
| **SYR + OH → RADSYR+ OH** | $9.63\ 10^{-11}$ |
| **RADSYR+ HO2 → 0.57 PSYR+ HO2** | $2.91\ 10^{-13}$exp(1300/T) |
| **RADSYR + NO → 0.36 PSYR+ NO** | $2.70\ 10^{-13}$exp(360/T) |
| **RADSYR + NO3 → 0.36 PSYR + NO3** | $2.30\ 10^{-12}$ |
| **GUAI + OH → RADGUAI+ OH** | $7.53\ 10^{-11}$ |
| **RADGUAI + HO2 → 0.37GHDPerox + HO2** | $2.91\ 10^{-13}$exp(1300/T) |
| **RADGUAI + NO → 0.32GHDPerox + NO** | $2.70\ 10^{-13}$exp(360/T) |
| **RADGUAI + NO3 → 0.32GHDPerox + NO3** | $2.30\ 10^{-12}$ |
| **NAPH + OH → NAPHP+ OH** | $2.44\ 10^{-11}$ |
| **NAPHP + HO2 → 0.44 BBPAHlN+ HO2** | $3.75\ 10^{-13}$ exp(980/T) |
| **NAPHP + MEO2 → 0.44 BBPAHlN+ MEO2** | $3.56\ 10^{-14}$ exp(708/T) |
| **NAPHP + C2O3 → 0.44 BBPAHlN+ C2O3** | $7.40\ 10^{-13}$ exp(765/T) |
| **NAPHP + NO → 0.26 BBPAHhN+ NO** | $2.70\ 10^{-11}$ exp(360/T) |
| **NAPHP + NO3 → 0.26 BBPAHhN+ NO3** | $1.2\ 10^{-12}$ |
| **MNAPH + OH → 0.26 MNAPHP+ OH** | $2.44\ 10^{-11}$ |
| **MNAPHP + HO2 → 0.46 BBPAHlN+ HO2** | $2.44\ 10^{-11}$ |
| **MNAPHP + MEO2 → 0.46 BBPAHlN+ MEO2** | $3.56\ 10^{-14}$ exp(708/T) |
| **MNAPHP + C2O3 → 0.46 BBPAHlN+ C2O3** | $7.40\ 10^{-13}$ exp(765/T) |





| | |
|---|---|
| **MNAPHP + NO → 0.37 BBPAHhN+ NO** | $2.70 \ 10^{-11} \exp(360/T)$ |
| **MNAPHP + NO3 → 0.37 BBPAHhN+ NO3** | $1.2 \ 10^{-12}$ |
| **USC>6$_{phen}$ + OH → 0.75 USC>6CAT + OH** | $4.7 \ 10^{-13} \exp(1220/T)$ |
| **USC>6CAT + OH → 0.28 USC>6ACIDMAL + OH** | $9.9 \ 10^{-10}$ |
| **USC>6$_{NAPH}$ + OH → USC>6NAPHP+ OH** | $2.44 \ 10^{-11}$ |
| **USC>6NAPHP + HO2 → 0.44 USC>6BBPAHlN+ HO2** | $3.75 \ 10^{-13} \exp(980/T)$ |
| **USC>6NAPHP + MEO2 → 0.44 USC>6BBPAHlN+ MEO2** | $3.56 \ 10^{-14} \exp(708/T)$ |
| **USC>6NAPHP + C2O3 → 0.44 USC>6BBPAHlN+ C2O3** | $7.40 \ 10^{-13} \exp(765/T)$ |
| **USC>6NAPHP + NO → 0.26 USC>6BBPAHhN+ NO** | $2.70 \ 10^{-11} \exp(360/T)$ |
| **USC>6NAPHP + NO3 → 0.26 USC>6BBPAHhN+ NO3** | $1.2 \ 10^{-12}$ |



**Appendix C**

**Table C1.** Ageing mechanism of I/S/L-VOCs using Couvidat approach (Couvidat et al., 2012).

$$BBPOAlP + OH \xrightarrow{k_a} BBSOAlP + OH \qquad \text{(CR1)}$$

$$BBPOAmP + OH \xrightarrow{k_a} BBSOAmP + OH \qquad \text{(CR2)}$$

$$BBPOAhP + OH \xrightarrow{k_a} BBSOAhP + OH \qquad \text{(CR3)}$$

With $k_a = 2.10^{-11} \ molecule^{-1}.cm^3.s^{-1}$

**Table C2.** Properties of primary and secondary I/S/L-VOCs.

| Surrogates | Emission fraction | Molecular weight (g.mol$^{-1}$) | log C$^*$ | Enthalpy of vaporization (kJ.mol$^{-1}$) |
|---|---|---|---|---|
| BBPOAlP | 0.25 | 280 | -0.04 | 106 |
| BBPOAmP | 0.32 | 280 | 1.94 | 91 |
| BBPOAhP | 0.43 | 280 | 3.51 | 79 |
| BBSOAlP | - | 392 | -2.04 | 106 |
| BBSOAmP | - | 392 | -0.06 | 91 |
| BBSOAhP | - | 392 | 1.51 | 79 |





**Appendix D**

**Table D1.** Ageing mechanism of I/S/L-VOCs using Ciarelli approach (Giancarlo et al., 2017).

$$BBPOA1 + OH \xrightarrow{\;k_b\;} BBSOA0 + OH \tag{DR4}$$

$$BBPOA2 + OH \xrightarrow{\;k_b\;} BBSOA1 + OH \tag{DR5}$$

$$BBPOA3 + OH \xrightarrow{\;k_b\;} BBSOA2 + OH \tag{DR6}$$

$$BBPOA4 + OH \xrightarrow{\;k_b\;} BBSOA3 + OH \tag{DR7}$$

10  $$BBSOA3 + OH \xrightarrow{\;k_b\;} BBSOA2 + OH \tag{DR8}$$

$$BBSOA2 + OH \xrightarrow{\;k_b\;} BBSOA1 + OH \tag{DR9}$$

$$BBSOA1 + OH \xrightarrow{\;k_b\;} BBSOA0 + OH \tag{DR10}$$

With $k_b = 4.10^{-11} \ molecule^{-1}.cm^3.s^{-1}$



**Table D2.** Properties of the VBS species (primary and secondary I/S/L-VOCs).

| Surrogates | Emission fraction | Molecular weight (g.mol$^{-1}$) | log C $^*$ | Enthalpy of vaporization (kJ.mol$^{-1}$) |
|---|---|---|---|---|
| BBPOA0 | 0.2 | 216 | -1 | 77.5 |
| BBPOA1 | 0.1 | 216 | 0 | 70 |
| BBPOA2 | 0.1 | 216 | 1 | 62.5 |
| BBPOA3 | 0.2 | 216 | 2 | 55 |
| BBPOA4 | 0.4 | 215 | 3 | 35 |
| BBSOA0 | - | 194 | -1 | 35 |
| BBSOA1 | - | 189 | 0 | 35 |
| BBSOA2 | - | 184 | 1 | 35 |
| BBSOA3 | - | 179 | 2 | 35 |





## Appendix E

**Table E1.** Summary of the parameters used to compute the dry-deposition velocities of the gaseous I/S/L-VOCs.

| Species | Molecular weight[a] | $C^{*\,b}$ | $H_{eff}{}^{c}$ | Reactivity fo | Diffusivity[d] | $\alpha^{e}$ | $\beta^{f}$ |
|---|---|---|---|---|---|---|---|
| BBPOAlP | 280 | 091 | $4.10^{+5}$ | 0.1 | 0.0634 | 0 | 0.05 |
| BBPOAmP | 280 | 87.09 | $1.6\ 10^{+5}$ | 0.1 | 0.0634 | 0 | 0.05 |
| BBPOAhP | 280 | 3235 | $10^{+5}$ | 0.1 | 0.0634 | 0 | 0.05 |
| BBSOAlP | 392 | 0.009 | $1.3\ 10^{+7}$ | 0.1 | 0.0388 | 0 | 0.5 |
| BBSOAmP | 392 | 0.87 | $4.\ 10^{+5}$ | 0.1 | 0.0388 | 0 | 0.5 |
| BBSOAhP | 392 | 32.35 | $1.45\ 10^{+5}$ | 0.1 | 0.0388 | 0 | 0.5 |
| BBPOA0 | 216 | 0.1 | $3.2\ 10^{+5}$ | 0.1 | 0.072 | 0 | 0.05 |
| BBPOA1 | 216 | 1 | $4\ 10^{+5}$ | 0.1 | 0.072 | 0 | 0.05 |
| BBPOA2 | 216 | 10 | $1.3\ 10^{+5}$ | 0.1 | 0.072 | 0 | 0.05 |
| BBPOA3 | 216 | 100 | $1.6\ 10^{+5}$ | 0.1 | 0.072 | 0 | 0.05 |
| BBPOA4 | 215 | 1000 | $10^{+5}$ | 0.1 | 0.072 | 0 | 0.05 |
| BBSOA0 | 194 | 0.1 | $3.2\ 10^{+5}$ | 0.1 | 0.0762 | 0 | 0.05 |
| BBSOA1 | 189 | 1 | $4.0\ 10^{+5}$ | 0.1 | 0.0771 | 0 | 0.05 |
| BBSOA2 | 184 | 10 | $1.3\ 10^{+5}$ | 0.1 | 0.0783 | 0 | 0.05 |
| BBSOA3 | 179 | 100 | $1.6\ 10^{+5}$ | 0.1 | 0.0793 | 0 | 0.05 |

[a] Molar weight (g.mol$^{-1}$)

5    [b] Saturation concentration ($\mu g \cdot m^{-3}$)

[c] Effective Henry constant (M.atm$^{-1}$)

[d] Diffusivity (cm$^{-2}$.s$^{-1}$)

[e] Parameter for curticle and soil resistance scaling to $SO_2$

[f] Parameter for curticle and soil resistance scaling to $O_3$



*Author contributions.* MM, KS, GL and FC developed the chemical mechanisms. ST and MM prepared VOC emissions from fires. MM performed the simulations, with help from MC and KS for the post-processing. MM, KS, GL prepared the manuscript with contributions from all co-authors.

*Acknowledgements.* CEREA is a member of the Institut Pierre-Simon Laplace (IPSL). A PhD grant from École des Ponts ParisTech funded
5    partially this research.



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
