# Peer review of "Precursors and formation of secondary organic aerosols from wildfires in the Euro-Mediterranean region"

_Atmospheric Chemistry and Physics, 2018_

## Referee Comment (RC1) · Anonymous Referee #1 · 16 Nov 2018

**General comments**

This study presents a new SOA formation mechanism developed to quantify the relative contribution of SOA precursors from wildfires in summer 2007 to organic aerosols in the Mediterranean region. The mechanism is an extension of an existing one by inclusion of some aromatic volatile organic compounds (VOC) emitted from wildfires. Since the wildfires have significant effects on the chemical composition of the atmosphere, a realistic representation of their emissions as well as their chemical fate in the models is important. Although I think such efforts might be valuable, this manuscript needs a major revision if accepted.

[Figure]

One of the weaknesses is that the manuscript presents simulations using an extended mechanism to quantify the relative contribution of precursors from wildfires to OA formation, but it does not provide any attempt to show how realistic the results are. It makes more sense to perform such studies during periods where detailed measurements –especially OA- are available to support the results, at least to a certain extent.

A general model evaluation (for both gaseous and particulate pollutants) to provide some confidence on the model performance during the selected period of time is the basis for all kind of modeling studies. Without such confidence it is impossible to get reasonable conclusions out of the simulations. In introduction, authors mention "a general good performance for PM2.5" citing another manuscript which is still under review.

Specific comments

1) The title indicates that the study is for the "Euro-Mediterranean" region. Results, however, mainly focus on a sub-region over Italian peninsula, Greece and some Balkan countries, named awkwardly as *MedReg". I assume the name comes from the definition of different regions in the Mediterranean used in Majdi et al. (2018) as MedReg1, MedReg2, etc., but it sounds strange when it stands alone in this manuscript. Authors might consider changing it, for example simply as "sub-region".

2) Page 1, line 23: Last sentence makes no sense.

3) Page 2, line 10-11: please replace "intermediate organic compounds" with "intermediate volatility organic compounds"

4) Page 12, Section 3: This section is too short. For the model set up authors cite Majdi et al. (2018) which is still under review. Even if that manuscript is accepted for publication, the modeling methods and detailed information about the model inputs must be described in this manuscript as well (meteorological parameters, anthropogenic and biogenic emissions (inventories, model, version), boundary conditions, deposition,

etc). The model domain covers an area where desert dust is very important. Some studies show significant dust contribution in the Mediterranean even below 2.5 micrometer (Fernandes et al., 2015; Denjean et al., 2016). Was dust included in the model simulations, in boundary conditions, how was dust distributed in model size fractions?

5) Page 12, line 27: Authors need to explain the reason of using CB05 mechanism by including additional compounds and reactions leading to SOA formation instead of using more advanced CB6 mechanisms (Yarwood et al., 2010) which have already some of these compounds.

6) Page 15, Section 5.1: Uncertainties in VOC emissions are probably very high. As I understand, authors considered only two types of vegetation (crop residue and chaparral) of which the emission factors were available and also assumed that temperate forest and savanna have the same EF as chaparral for cresol, catechol, guaiacol, syringol, naphthalene and methylnaphtalene. Additional discussion about the variability of emissions from different vegetation with references is needed to justify this assumption.

7) It is known that terpenoid emissions –especially monoterpenes- increase during forest fires (Ciccioli et al., 2014). Since monoterpenes are the essential precursors for SOA formation, their contribution to OA during wildfires might be larger when taken into account in addition to their natural emissions. Although not mentioned in the manuscript, I assume MEGAN model was used to estimate the BVOC emissions. At least some discussion about the contribution of the increased BVOC emissions to SOA formation during wildfires might be useful.

8) Page 13, Fig. 5: I assume that the MedReg is not another nested domain (same resolution as the red dotted domain). It has to be explained better in the text the choice of the green box named as MedReg -which is odd

9) Page 25, Conclusions are very short, based only on model calculations without any supporting material or discussions. This section needs a revision.

10) Please change Giancarlo et al., 2017 to Ciarelli et al., 2017 in page 3, line 16, 29, 33, page 12, line 13, page 13, line 15, page 16, line 30, page 33, line 2

11) Page 37, line 25: Please correct "Lowik, J." as "Slowik J."

12) Page 38, line 27: Please correct the following reference: "Giancarlo, C.and El Hadad, I., Bruns, E., Aksoyoglu, S., Mohler, O., Baltensperger, U., and Prevot, A.: Constraining a hybrid volatility basis set model for aging wood burning emissions using smog chamber experiments, Geosci. Model Dev., 10, 2303–2320, https://doi.org/10.5194/gmd-10-2303-2017, 2017" as "Ciarelli, G., El Haddad, I., Bruns, E., Aksoyoglu, S., Möhler, O., Baltensperger, U., and Prévôt, A. S. H.: Constraining a hybrid volatility basis-set model for aging of wood-burning emissions using smog chamber experiments: a box-model study based on the VBS scheme of the CAMx model (v5.40), Geosci. Model Dev., 10, 2303-2320, 10.5194/gmd-10-2303-2017, 2017"

References

Ciccioli, P., Centritto, M., and Loreto, F.: Biogenic volatile organic compound emissions from vegetation fires, Plant, Cell & Environment, 37, 1810-1825, doi:10.1111/pce.12336 (2014).

Denjean, C., Cassola, F., Mazzino, A., Triquet, S., Chevaillier, S., Grand, N., Bourrianne, T., Momboisse, G., Sellegri, K., Schwarzenbock, A., Freney, E., Mallet, M., and Formenti, P.: Size distribution and optical properties of mineral dust aerosols transported in the western Mediterranean, Atmos. Chem. Phys., 16, 1081-1104, 10.5194/acp-16-1081-2016, (2016).

A. P. Fernandes, M. Riffler, J. Ferreira, S. Wunderle, C. Borrego and O. Tchepel, Comparisons of aerosol optical depth provided by Severi satellite observations and CAMx air quality modeling, The International Archives of the Photogrammetry, Remote Sensing and Spatial Information Sciences, Volume XL-7/W3, 2015, 36th International Symposium on Remote Sensing of Environment, 11–15 May 2015, Berlin, Germany

[Figure]

Yarwood, G., J. Jung, G. Z. Whitten, G. Heo, J. Mellberg and E. Estes. Updates to the Carbon Bond Mechanism for Version 6 (CB6). Presented at the 9th Annual CMAS Conference, Chapel Hill, October (2010).

---

## Referee Comment (RC2) · Anonymous Referee #2 · 27 Nov 2018

Comments on 'Precursors and formation of secondary organic aerosols from wildfires in the Euro-Mediterranean region', Majdi, et al., (2018)

Summary/recommendation:
This paper was an interesting modelling study on capturing both precursor vapors and SOA from wildfires. The authors updated a pre-existing SOA formation scheme in order to run several sensitivity simulations that detail the sensitivity of their model to precursor sources and concentrations as well as aging mechanisms on smoke OA. However, many portions of the paper were made difficult to follow due to missing (mainly minor) details. I recommend that this study be published but with minor revisions. I request that the authors consider the following points as they revise this manuscript:

General comments:
Pg 2, line 15: the definition of $OA_{tot}$ is confusing, given that 'aerosol' usually refers to the particle phase concentrations only. If this is the sum of the particle and gas phase, do the authors mean that the gas phase species are only those who are low enough in volatility to participate in partitioning? Or all the gas phase species, including VOCs and IVOCs? Please clarify this.

pg 3, line 27 Please qualify what is meant by "misclassified" here

Section 2.1: the authors should consider adding more explanation as to what the original $H^2O$ scheme was (and what its purpose is), and what additions/changes the authors are specifically making to $H^2O$. It's a little unclear if the details being described on pg 4 through line 6 on pg 5 are of the original $H^2O$ model? The first 2 sentences of this section (starting on pg 4, line 18) would benefit from having the appropriate $H^2O$ citations added.

Pg 5, lines 23-24. The authors state that the one-product model correctly reproduces the experimental data; there is a small amount of spread between the model and experimental data. Can the authors briefly quantify that error? Same for the analysis given for fig 2 (lines 16-17 of pg 6)

Pg 6, lines 1-2: can the authors briefly discuss what error might be anticipated to be introduced by using ACIDMAL as a high-NOx surrogate given the lack of data for this mechanism? Same for the cresol chemical mechanism, lines 13-15 of page 6?

Pg 11 lines 13-16: can the authors briefly explain their rationale for choosing USC>6 compounds to undergo the same OH oxidation mechanisms as phenol or naphthalene?

Section 2.2: The acronyms should be well defined: what are BBPOAlP, BBPOAmP and BBPOAhP? I strongly suggest making sure all acronyms in this work are well-defined the first time they are used. Also, consider re-defining major (uncommon) acronyms at the beginning of new sections for any readers who may be skipping sections. These aren't defined to my knowledge until section 5.2. 'P' is never defined that I saw--pressure? There is a missing citation or statement on line 6 (currently shows up as a questions mark). Also, it should be made clear in the text to which volatility bin BBPOA0, BBPOA1, etc belongs to.

Pg 12 lines 15-18: It's not clear from the text or appendix D what the fragmentation and functionalization scheme is. It would be helpful to have the fragmentation and functionalization rates or fractions explicitly expressed. Or is the given reaction rate with OH of $4e10^{-11}$ supposed to account a combined probability of fragmentation and functionalization? The units on this reaction rate seem incorrect, they are listed as molecules$^{-1}$ cm$^3$ s$^{-1}$, where often reaction rates are expressed as molecules cm$^{-3}$ s$^{-1}$. Please comment on the units. Also, a brief look through Donahue et al. (2013) does not show where the specific value of $4e10^{-11}$ came from--perhaps another citation is also necessary here? Can the authors comment on this as well. Finally, it should be stated what happens to fragmentation products--are they placed into higher volatility bins or are they "lost" and no longer tracked in the model? The authors should consider adding more details on all of the issues raised here in the text.

Section 3 lines 30-31: I suggest writing out what ISORROPIA and SOAP stand for.

Page 13, line 6 and Table E1: I suggest adding 1 sentence explanation of what the reactivity factor is. In Table E1 this is listed as Reactivity fo, consider changing to something like Reactivity factor (fo).

Section 4 page 13 line 12: It would be helpful to let the reader know that the emissions estimate of toluene and xylene will be discussed in the next section. Same for when NMOG is discussed in this section.

Pg 14: What is Un in the Multstep-UnNMOG-withVOCs?

pg 14 lines 8-12: can the authors comment on by ~ how much (I assume a range) lower Donahue et al (2005)'s calculation of the enthalpy of vaporization was than the SIMPOL.1 calculations? I recommend including the range of delta($H_{vap}$)s from SIMPOL.1 either in the text or in table 1.

Section 5.1: can the authors comment on how representative they believe woodfire stove smoke emissions are of wildfires?

Section 5.3, lines 32-33 (first sentence of the section): would the left panel of Fig 7 technically be showing $POA_{tot}$? Since these are the $OA_{tot}$ precursors?

Section 6.1. Pg 21 lines 3-5: Can the authors briefly justify the choice of using Multstep-withVOCs for this figure?

Page 21 lines 9-10: from which model run(s) does this data come from?

Pg 23, lines 10-13: how were the differences within "the fire plume" determined? What's meant by the fire plume here? How well can the model resolve an individual plume? Please explain this further.

Two more general comments: Are the model results being compared to actual observations? If instead they are being compared to work done in the first author's other ACPD paper, this should be made more clear and the comparisons could be spelled out more explicitly.

This work would benefit from a discussion of the pros/cons of each model simulation type, and whether or not any model simulations appear to better represent the real atmosphere. Much work was clearly done here, but the paper currently does not seem to have the "why this matters/how it improves upon previous work" factor yet that will allow it to become an easily useful guide and reference for other members of the community.

Figures/tables:
Figures 1-4 would benefit from being made in a higher-quality format rather than the excel default graphs.

Figure 11: It should state in the figure caption and/or on the figure which model simulation is being used to make this figure.

Figure 12: from what data/model simulation(s) was this pie chart constructed? This should be stated in the figure caption and in the text.

Figure 13: the colorbars should have units with them (% and mass concentrations?). This colorbar is a little hard to interpret, are we to take that the tan regions are anywhere between 0-5 or 8% different? Can the authors make the colorbars for each % difference plot the same, they're currently changing by between 5 and 11 units. I suggest considering a non-linear colorbar to see more structure within the -5 to 5% difference range.

Technical comments:

Page 2 line 7: suggest rewriting to PM is composed of organic and inorganic compounds, dust, and black carbon (Jimenez et al., 2009).

Page 2 line 13: do the authors mean that both POA and SOA are composed of compounds of different volatilities? Suggest clarifying this sentence.

Page 20 line 11: un should be Un in the simulation name.

Page 20 line 21: this would make more sense if it was written something like "Across our cases, 28 to 42%..."

---

## Author Comment (AC1) · 15 Feb 2019

The authors wish to thank the anonymous referee for the very helpful comments and corrections. All corrections have been included in this new version. A response to the general and specific comments is provided below (in blue).

**General comments :**

**This study presents a new SOA formation mechanism developed to quantify the relative contribution of SOA precursors from wildfires in summer 2007 to organic aerosols in the Mediterranean region. The mechanism is an extension of an existing one by inclusion of some aromatic volatile organic compounds (VOC) emitted from wildfires. Since the wildfires have significant effects on the chemical composition of the atmosphere, a realistic representation of their emissions as well as their chemical fate in the models is important. Although I think such efforts might be valuable, this manuscript needs a major revision if accepted.**

**One of the weaknesses is that the manuscript presents simulations using an extended mechanism to quantify the relative contribution of precursors from wildfires to OA formation, but it does not provide any attempt to show how realistic the results are. It makes more sense to perform such studies during periods where detailed measurements –especially OA- are available to support the results, at least to a certain extent.**
**A general model evaluation (for both gaseous and particulate pollutants) to provide some confidence on the model performance during the selected period of time is the basis for all kind of modeling studies. Without such confidence it is impossible to get reasonable conclusions out of the simulations. In introduction, authors mention "a general good performance for PM2.5" citing another manuscript which is still under review.**

Since the lack of available organic surface data close to the fire region during the summer 2007, we could not evaluate the organic gaseous and organic particulate pollutants which is in fact a limitation of our work. The paper of Majdi et al. (2019) is now published in ACP. It evaluates the simulated PM2.5 concentrations during the summer 2007 by comparison to available measurements of PM2.5 concentrations and aerosol optical properties. The general performances of the model for the PM2.5 concentrations during the summer 2007 are good, although they are slightly underestimated, compared to surface measurements at 8 AIRBASE stations. Besides, the new SOA formation mechanism developed in this work is based on measurement studies from smog chamber experiments. This gives confidence in the results,

although we agree that some further validation with organic concentrations close to fires is required.

**Specific comments :**

**1) The title indicates that the study is for the "Euro-Mediterranean" region. Results, however, mainly focus on a sub-region over Italian peninsula, Greece and some Balkan countries, named awkwardly as *MedReg". I assume the name comes from the definition of different regions in the Mediterranean used in Majdi et al. (2018) as MedReg1, MedReg2, etc., but it sounds strange when it stands alone in this manuscript. Authors might consider changing it, for example simply as "sub-region".**

In this work, we chose to focus more on MedReg which is considered to be the most affected subregion by fire according to Majdi et al. (2019).
Taking into account the referee's comment, MedReg is replaced by subregion in the revised version of the paper.

**2) Page 1, line 23: Last sentence makes no sense.**

The sentence in page 1, line 23 « Considering the VOC emissions results in a moderate increase of PM2.5 concentrations mainly in Balkans (up to 24%) and in the fire plume (+10%). " is replaced by "Considering VOCs as SOA precursors results in a moderate increase of PM2.5 concentrations mainly in Balkans (up to 24%) and in the fire plume (+10%). »

**3) Page 2, line 10-11: please replace "intermediate organic compounds" with "intermediate volatility organic compounds"**

« Intermediate organic compounds » in page 2, line 10-11 is replaced by « intermediate volatility organic compounds » in the revised version of the paper.

**4) Page 12, Section 3: This section is too short. For the model set up authors cite Majdi et al. (2018) which is still under review. Even if that manuscript is accepted for publication, the modeling methods and detailed information about the model inputs must be described in this manuscript as well (meteorological parameters, anthropogenic and biogenic emissions (inventories, model, version), boundary conditions, deposition, etc).**

A paragraph describing the model set up is added in page 12 line 23 in the revised version of the paper : « Here, the CTM Polair3D/Polyphemus (Mallet et al., 2007; Sartelet et al., 2012) is used with a similar set up as described in Majdi et al. (2019)

and summarized here.
A modified version of the Carbon Bond 05 model (CB05) (Yarwood et al., 2005, Kim et al., 2011) is used for gas-phase chemistry with the SIze REsolved Aerosol Model (SIREAM) (Debry et al., 2007) for aerosol dynamics (coagulation, condensation/evaporation). The meteorological fields are provided by the European Center for Medium-Range Weather Forecast (ECMWF, ERA-Interim). Boundary conditions of the nesting domain are obtained from the global chemistry-transport model MOZART-GEOS5 6 hourly simulations outputs (Emmons et al. , 2010). Anthropogenic emissions are generated from EMEP inventory for 2007 (European Monitoring and Evaluation Program, http://www.emep.int). Biogenic emissions are estimated with the Model of emissions of Gases and Aerosols for Nature (MEGAN-LHIV, Guenther et al., 2006). Sea-salt emissions are parameterized following Monahan (1986). Soil and surface database of Menut et al. (2013) is used to calculate the dust emissions considering the spatial extension of potentially emitted area in Europe described in Briant et al. (2017). The daily fire emissions are calculated using the APIFLAME fire emissions model v 1.0 (Turquety et al., 2014) as described in Majdi et al. (2019). »

**5) The model domain covers an area where desert dust is very important.**
**Some studies show significant dust contribution in the Mediterranean even below 2.5 micrometer (Fernandes et al., 2015; Denjean et al., 2016). Was dust included in the model simulations, in boundary conditions, how was dust distributed in model size fractions?**

Dust was included in the model simulations, as now described in the model description (see reply to comment 4). Boundary conditions from the domain studied were obtained from the simulation on the larger domain (nesting domain).
Dust was distributed into 4 diameter bins (between 0.01 μm and 10 μm).

**5) Page 12, line 27: Authors need to explain the reason of using CB05 mechanism by including additional compounds and reactions leading to SOA formation instead of using more advanced CB6 mechanisms (Yarwood et al., 2010) which have already some of these compounds.**

CB05 and CB06 do not treat the SOA formation, they consider some SOA precursors, but the oxidation of the SOA precursors do not form semi-volatile organic compounds that can condense onto particles. In this work, we used a version of CB05 which was modified by Couvidat et al. (2012) and Kim et al. (2011) to integrate the formation of semi-volatile organic compounds from some VOCs that are SOA precursors. The purpose of this work is to develop further this modified version of CB05 to integrate the formation of semi-volatile organic compounds from more VOCs that are SOA precursors.

**6) Page 15, Section 5.1: Uncertainties in VOC emissions are probably very high. As I understand, authors considered only two types of vegetation (crop residue and chaparral) of which the emission factors were available and also assumed that temperate forest and savanna have the same EF as chaparral for cresol, catechol, guaiacol, syringol, naphthalene and methylnaphthalene. Additional discussion about the variability of emissions from different vegetation with references is needed to justify this assumption.**

We assume that temperate forest and savanna have the same EF as chaparral for cresol, catechol, guaiacol, syringol, naphthalene and methylnaphthalene. This assumption is made to not underestimate their emissions since temperate and savanna are the most burned vegetation in the Euro-Mediterranean region according to Akagi et al. (2011).

As described in the paper « Because the EF of VOCs emitted by wildfires of crop residue, chaparral, temperate forest and savanna in the inventory of Akagi et al. (2011) are often of the same order of magnitude (Table A1 of Appendix A), it is assumed here that temperate forest and savanna have the same EF as chaparral for cresol, catechol, guaiacol, syringol, naphthalene and methylnaphthalene. »

A discussion about uncertainties and the variability of emissions from different vegetation is added after page 16 line 12 :

« This assumption is justified by considering uncertainties linked to emissions : Turquety et al. (2014) estimated that the uncertainties on the emitted carbon related to fire emissions can reach 100%. They found that the database used for the type of vegetation burned plays a significant role on the emitted carbon (~ 75% associated uncertainty). Moreover, the inventory used in this work (APIFLAME (Turquety et al., 2014))  is mainly based on the emission factors of Akagi et al. (2011) using data from different field and laboratory experiments. Uncertainties related to these emissions factors are high. For example, Alves et al. (2011) measured carbon monoxide (CO) emissions for forest fires in Portugal 2.6 times higher than the values of Akagi et al. (2011) for extra-tropical forests. »

**7) It is known that terpenoid emissions –especially monoterpenes- increase during forest fires (Ciccioli et al., 2014). Since monoterpenes are the essential precursors for SOA formation, their contribution to OA during wildfires might be larger when taken into account in addition to their natural emissions. Although not mentioned in the manuscript, I assume MEGAN model was used to estimate the BVOC emissions. At least some discussion about the contribution of the increased BVOC emissions to SOA formation during wildfires might be useful.**

Yes, MEGAN is used (see reply to comment 4). Majdi et al. (2019) highlighted that during the  fire episodes in summer 2007,  SOA and POA from I/S/L-VOCs are the major PM2.5 component , their contribution to PM2.5 concentrations is larger than the SOA from biogenics (57% vs 0.3%). However, the potential increase of terpenoid emissions by forest fires is now specified in section 5.1 after line 11 page 15 : « Note

that although Biogenic VOC (BVOC) emissions may increase during wildfires, as suggested by Cicciolo et al. (2014), the potential increase of BVOC emissions from wildfires is not considered here due to lack of data. »

**8) Page 13, Fig. 5: I assume that the MedReg is not another nested domain (same resolution as the red dotted domain). It has to be explained better in the text the choice of the green box named as MedReg -which is odd.**

Indeed, MedReg is not a nested domain, it is the subregion where the most severe fire episodes occur according to Majdi et al. (2019).
The sentence « Since the largest fires in the Euro-Mediterranean domain occur mainly in Balkan and Eastern Europe (between 20 July and 31 July 2007), in Greece (between 24 August and 30 August) and in Southern Italy (between 9 July and 31 July 2007 ) (Majdi et al., 2019), we choose to focus on the subregion indicated in green box in Figure 5. » is added to page 12 line 26 in the new version of the paper.

**9) Page 25, Conclusions are very short, based only on model calculations without any supporting material or discussions. This section needs a revision.**

The conclusion was modified to better stress out how this work improves upon previous work, and how it can be useful for other members of the community.

« This study quantified the relative contribution of OAtot precursors (VOCs, I/S/L-VOCs) emitted by wildfires to OA formation and particle concentrations, during the summer 2007 over the Euro-Mediterranean region. A new chemical mechanism H2Oaro was developed to represent the SOA formation from selected VOCs, namely toluene, xylene, benzene, phenol, cresol, catechol, furan, guaiacol, syringol, naphthalene, methylnaphthalene, the structurally assigned and unassigned compounds with at least carbon atoms per molecule (USC>6), based on smog chamber experiments under low and high-NOx conditions. This mechanism was implemented in the chemistry transport model Polair3D of the air-quality platform Polyphemus. Over the Euro-Mediterranean area, the OA concentrations emitted by wildfires originate mostly from I/S/L-VOCs. The OA concentrations from gaseous I/S/L-VOCs are about 10 times higher than the OA concentrations from VOCs. However, the contribution of the oxidation of VOCs to the OA concentrations is locally significant (it reaches 30% close to the area where wildfires are emitted and 20% in the fire plume). Air-quality models often represent SOA formation from only a few VOCs, such as toluene and xylene. This study points out the need to consider the contribution of a variety of VOCs, namely, phenol, benzene, catechol, cresol, xylene, toluene and syringol, when modelling SOA formation from wildfires. The contribution of these VOCs may even be underestimated here for two reasons. First, the yields from smoke chamber experiments were not corrected for wall losses, and they may therefore be underestimated leading to an underestimation of the SOA formation from VOCs in the model. Second, a large part of OA concentrations from

VOCs is in the gas phase (› 70%). This suggests that the influence of the VOC emissions on OA concentrations could be larger, if the surrogates from these VOC oxidations partition more easily to the particle phase. This could be the case if further ageing mechanisms are considered for these VOCs or if the particles are very viscous (Kim et al., 2019).

Emissions of gaseous I/S/L-VOCs are a large source of uncertainties. However, similar estimates were obtained here by using as a proxy POA emissions (with a factor of 1.5) or NMOG emissions (with a factor of 0.36). Sensitivity simulations were performed to quantify the uncertainties on OA and PM2.5 concentrations linked to I/S/L-VOCs emissions and chemical evolution (ageing). They are found to be lower than the uncertainties associated with SOA formation from VOC emissions. This stresses the need to consider a variety of VOCs in SOA formation model, and to better characterize their emission factors. »

**10) Please change Giancarlo et al., 2017 to Ciarelli et al., 2017 in page 3, line 16, 29, 33, page 12, line 13, page 13, line 15, page 16, line 30, page 33, line 2**

Giancarlo et al., 2017  is replaced by  Ciarelli et al., 2017 in  the new version of the paper.

**11) Page 37, line 25: Please correct "Lowik, J." as "Slowik J."**

Lowik, J. In page 37 line 25 is corrected to Slowik, J..

**12) Page 38, line 27: Please correct the following reference: "Giancarlo, C.and El Hadad, I., Bruns, E., Aksoyoglu, S., Mohler, O., Baltensperger, U., and Pre- vot, A.: Constraining a hybrid volatility basis set model for aging wood burning emissions using smog chamber experiments, Geosci. Model Dev., 10, 2303–2320, https://doi.org/10.5194/gmd-10-2303-2017, 2017" as "Ciarelli, G., El Haddad, I., Bruns, E., Aksoyoglu, S., Möhler, O., Baltensperger, U., and Prévôt, A. S. H.: Constraining a hybrid volatility basis-set model for aging of wood-burning emissions using smog cham- ber experiments: a box-model study based on the VBS scheme of the CAMx model (v5.40), Geosci. Model Dev., 10, 2303-2320, 10.5194/gmd-10-2303-2017, 2017"**

The reference in page 38, line 27 is corrected.

**References:**

Ciccioli, P., Centritto, M., and Loreto, F.: Biogenic volatile organic compound emissions from vegetation fires, Plant, Cell & Environment, 37, 1810-1825, doi:10.1111/pce.12336 (2014).
Denjean, C., Cassola, F., Mazzino, A., Triquet, S., Chevaillier, S., Grand, N., Bour-rianne, T., Momboisse, G., Sellegri, K., Schwarzenbock, A., Freney, E., Mallet, M.,

and Formenti, P.: Size distribution and optical properties of mineral dust aerosols transported in the western Mediterranean, Atmos. Chem. Phys., 16, 1081-1104, 10.5194/acp-16-1081-2016, (2016).

A. P. Fernandes, M. Riffler, J. Ferreira, S. Wunderle, C. Borrego and O. Tchepel, Com- parisons of aerosol optical depth provided by Severi satellite observations and CAMx air quality modeling, The International Archives of the Photogrammetry, Remote Sens- ing and Spatial Information Sciences, Volume XL-7/W3, 2015, 36th International Sym- posium on Remote Sensing of Environment, 11–15 May 2015, Berlin, Germany

Yarwood, G., J. Jung, G. Z. Whitten, G. Heo, J. Mellberg and E. Estes. Updates to the Carbon Bond Mechanism for Version 6 (CB6). Presented at the 9th Annual CMAS Conference, Chapel Hill, October (2010).

---

## Author Comment (AC2) · 15 Feb 2019

**Comments on 'Precursors and formation of secondary organic aerosols from wildfires in the Euro-Mediterranean region', Majdi, et al., (2018)**

**Anonymous Referee #2**

The authors wish to thank the anonymous referee for the very helpful comments and corrections. All corrections have been included in this new version. A response to the general and specific comments is provided below (in blue).

**General comments:**

**Pg 2, line 15: the definition of OAtot is confusing, given that 'aerosol' usually refers to the particle phase concentrations only. If this is the sum of the particle and gas phase, do the authors mean that the gas phase species are only those who are low enough in volatility to participate in partitioning? Or all the gas phase species, including VOCs and IVOCs? Please clarify this.**

We used the notation of Murphy et al. (2014), as now specified. In this notation, OAtot means the sum of particle and gas phase organic compounds of volatility lower than VOCs (i.e. of saturation concentration lower than $C*=3.2 \times 10^6$ µg.m$^{-3}$).

The sentence in page 2, line 15 is replaced in the revised version of the paper by:

« In the following, following Murphy et al. (2014), OAtot denotes the sum of gaseous and particle phase organic aerosol concentrations of volatility lower than VOCs. »

**pg 3, line 27 Please qualify what is meant by "misclassified" here**

For clarity, the sentence « Although primary gaseous I/S/L-VOCs are not considered or misclassified in emissions inventories » is replaced by « Although primary gaseous I/S/L-VOCs are not considered or classified as unspeciated NMOG in emissions inventories »

**Section 2.1: the authors should consider adding more explanation as to what the original H²O scheme was (and what its purpose is), and what additions/changes the authors are specifically making to H²O. It's a little unclear if the details being described on pg 4 through line 6 on pg 5 are of the original H²O model?**

**The first 2 sentences of this section (starting on pg 4, line 18) would benefit from having the appropriate H²O citations added.**

The sentence « The new mechanism (H2Oaro) is an extension of hydrophilic/hydrophobic organic (H2O) SOA mechanism » is replaced by « The new mechanism (H2Oaro) is an extension of the hydrophilic/hydrophobic organic (H2O) SOA mechanism, which details the formation of organic aerosols from the oxidation of precursors (Couvidat et al. 2012). Laboratory chamber studies provide the fundamental data that are used to parameterize the atmospheric SOA formation under low/high-NOx conditions. The formed organic aerosols are represented by surrogate compounds, with varying water affinity (hydrophobic, hydrophilic). In the original H2O mechanism, the precursors are I/S/L-VOCs, aromatics (xylene and toluene), isoprene, monoterpenes, sesquiterpene. In the extention H2Oaro developped here, other VOCs are considered as SOA precursors (phenol, cresol, catechol, benzene, furan, guaiacol, syringol, naphthalene, methylnaphthalene).»

Because the list of VOCs is now detailed in the description of H2Oaro, the first sentence of section 2.1 is simplified and the sentence « This section presents a new SOA formation mechanism H2Oaro developed to represent the SOA formation from the main VOCs that are estimated to be SOA precursors (phenol, cresol, catechol, benzene, furan, guaiacol, syringol, naphthalene, methylnaphthalene). » is replaced by « This section presents a new SOA formation mechanism H2Oaro developed to represent the SOA formation from the main aromatic VOCs that are estimated to be SOA precursors . »

**Pg 5, lines 23-24. The authors state that the one-product model correctly reproduces the experimental data; there is a small amount of spread between the model and experimental data. Can the authors briefly quantify that error? Same for the analysis given for fig 2 (lines 16-17 of pg 6)**

To quantify the small amount of spread between the model and experimental data, we calculate the RMSE as follows:

RMSE (%) = 100 x $(\sum (Yield_{exp} - Yield_{model})^2 / N)^{1/2}$

$Yield_{model}$ : The modeled SOA yield

$Yield_{exp}$: The experimental SOA yield

N: number of experiments

For figure 1 page 6, RMSE=3.1%

For figure 2 page 7, RMSE= 2.87%

This is added in page 5 lines 23-24 in the revised version of the paper as follows : « The one-product model with a stoechiometric coefficient α1 of 0.28 and a vapor pressure of 4.59 10−8 torr correctly reproduces the experimental data wih a  small amount of spread between the model and experimental data (RMSE of 3.1%). » and in page 6 lines 16-17 : « Figure 2 plots the SOA yields against the SOA concentrations. A stoechiometric coefficient and a saturation vapor pressure 0.39 and 3.52 10−6 torr respectively are found to fit accurately the experimental data with small differences between the model and experimental data (RMSE of ~ 3 % ) .»

**Pg 6, lines 1-2: can the authors briefly discuss what error might be anticipated to be introduced by using ACIDMAL as a high-NOx surrogate given the lack of data for this mechanism? Same for the cresol chemical mechanism, lines 13-15 of page 6?**

In this work, for catechol and cresol, we did not differentiate low-NOx and high-NOx oxidation, because of the lack of data for high-NOx conditions. Because of the lack of data, it is difficult to estimate what is the error associated to this assumption.

**Pg 11 lines 13-16: can the authors briefly explain their rationale for choosing USC>6 compounds to undergo the same OH oxidation mechanisms as phenol or naphthalene?**

This assumption is based on the results of the smog chamber of Bruns et al. (2016) : they quantified the SOA yield from USC>6 and found that their yields are significant. However, because their OH oxidation mechanism may not be easily defined, we chose to represent it with a compound which also has high yields. Phenol and naphtalene are good candidates. Because the oxidation products of naphtalene and phenol are very different (e.g. volatility), a sensitivity simulation is performed on choosing the oxidation mechanism of naphtalene rather than phenol, to evaluate the impact of the changing the oxidation mechanism.

Page 11, line 15, the following sentence is removed : « In this study, USC>6 compounds are assumed to undergo either the same OH oxidation mechanisms as phenol or as naphthalene, which are previously discussed in sections 2.1.1 and 2.1.6 respectively. », and it is replaced by the following sentences: « Because Bruns et al. (2016) estimated that SOA yields for USC>6 compounds are high, they are represented in the model by a high-yield compound. Phenol and naphtalene are good candidates. Because the oxidation products of naphtalene and phenol are very different (e.g. volatility), a sensitivity simulation is performed on choosing the oxidation mechanism of naphtalene rather than phenol, to evaluate the impact of changing the oxidation mechanism. »

**Section 2.2: The acronyms should be well defined: what are BBPOAlP, BBPOAmP and BBPOAhP? I strongly suggest making sure all acronyms in this work are well-defined the first time they are used. Also, consider re-defining major (uncommon) acronyms at the beginning of new sections for any readers who may be skipping sections. These aren't defined to my knowledge until section 5.2. 'P' is never defined that I saw--pressure? There is a missing citation or statement on line 6 (currently shows up as a questions mark). Also, it should be made clear in the text to which volatility bin BBPOA0, BBPOA1, etc belongs to.**

The sentences in page 12 line 3 are replaced in the revised version of the paper by:

« The primary organic aerosols emitted by biomass burning (BBPOAlP for compounds of low volatility, BBPOAmP for compounds of medium volatility and BBPOAhP for compounds of high volatility, of saturation concentration C*: log(C*)= -0.04, 1.93, 3.5 respectively) undergo one oxidation step in the gas phase, leading to the formation of secondary surrogates (BBSOAlP, BBSOAmP and BBSOAhP). »

The missing citation is added to line 6 page 12 as follows: « In the one-step oxidation scheme, used for example in Couvidat et al. (2012); Zhu et al. (2016); Sartelet et al. (2018) ... »

A reference to the volatility bins of the compounds BBPOA0, BBPOA1 etc are added page 12, line 15 : « BBPOA0, BBPOA1, BBPOA2, BBPOA3, BBPOA4 refer to the primary surrogates and BBSOA0, BBSOA1, BBSOA2, BBSOA3 refer to the secondary ones (see Table D2 of Appendix D for their properties). »

**Pg 12 lines 15-18: It's not clear from the text or appendix D what the fragmentation and functionalization scheme is. It would be helpful to have the fragmentation and functionalization rates or fractions explicitly expressed. Or is the given reaction rate with OH of 4e10- 11 supposed to account a combined probability of fragmentation and functionalization?**

**The units on this reaction rate seem incorrect, they are listed as molecules- 1 cm3 s- 1, where often reaction rates are expressed as molecules cm- 3 s- 1.**

**Please comment on the units.**

**Also, a brief look through Donahue et al. (2013) does not show where the specific value of 4e10- 11 came from--perhaps another citation is also necessary here? Can the authors comment on this as well.**

**Finally, it should be stated what happens to fragmentation products--are they placed into higher volatility bins or are they "lost" and no longer tracked in the model? The authors should consider adding more details on all of the issues raised here in the text.**

The unit of the reaction rate are molecule$^{-1}$ cm$^3$ s$^{-1}$ because it is a second order reaction rate. Indeed, the reaction takes into account a combined probability of

fragmentation and functionalization, which are considered simultaneously in each oxidation reaction.

The experimental reaction rate of $4e10^{-11}$ molecule$^{-1}$ cm$^3$ s$^{-1}$ is from Robinson et al. (2007). The sentences in page 12 line 16 is modified in the revised version as follows: « In the gas phase, the primary and secondary surrogates react with OH at a rate of $4.10^{-11}$ molecules$^{-1}$ cm$^3$ s$^{-1}$ (Robinson et al., 2007). »

High volatility fragmentation products are not considered in the parameterisations. Since fragmentation and functionalization are considered simultaneously in each oxidation reaction, the oxidation products correspond to fragmentation and funtionalization products which are placed into lower volatility bins than the precursor.

The sentences in page 12 lines 17-18 are modified as follows: « During each oxidation step, the oxidation of the surrogate increases the surrogate oxygen number and decreases its volatility and carbon number, due to functionalization and fragmentation which are considered simultaneously during each oxidation reaction. »

**Section 3 lines 30-31: I suggest writing out what ISORROPIA and SOAP stand for.**

The full names are added in the revised version of the paper as follows:

ISORROPIA refers to a thermodynamic equilibrium model for multiphase multicomponent inorganic aerosols.

SOAP stands for Secondary Organic Aerosol Processor.

**Page 13, line 6 and Table E1: I suggest adding 1 sentence explanation of what the reactivity factor is. In Table E1 this is listed as Reactivity fo, consider changing to something like Reactivity factor (fo).**

This sentence in page 13 line 6 is modified as follows: « The reactivity factor (f0), which corresponds to the ability of a dissolved gas to oxidize biological substances in solution, may range from 0 for non-reactive species to 1 for highly reactive species. In this work, the f0 value is set to 0.1 (Karl et al., 2010; Knote et al., 2015). »

Reactivity fo in Table 1 is replaced by Reactivity factor (f0).

**Section 4 page 13 line 12: It would be helpful to let the reader know that the emissions estimate of toluene and xylene will be discussed in the next section. Same for when NMOG is discussed in this section.**

The sentence in page 13 line 11-12 is modified in the revised version of the paper as follows: « for VOC emissions, only toluene and xylene are considered (as detailed in section 5.1) ... »

The sentence in page 14 line 2-3 is modified in the revised version of the paper as

follows: « but the gaseous I/S/L-VOC emissions are calculated from NMOG ( as described in section 5.2) ... »

**Pg 14: What is Un in the Multstep-UnNMOG-withVOCs?**

In the Multstep-UnNMOG-withVOC, Un stands for unidentified NMOG.

For clarity, the sentence page 14 line 3 « but the gaseous I/S/L-VOC emissions are

calculated from NMOG » is modified to « but the gaseous I/S/L-VOC emissions are assumed to be unidentified NMOG and they are estimated from NMOG emissions.»

**pg 14 lines 8-12: can the authors comment on by ~ how much (I assume a range) lower Donahue et al (2005)'s calculation of the enthalpy of vaporization was than the SIMPOL.1 calculations? I recommend including the range of delta(H vap) s from SIMPOL.1 either in the text or in table 1.**

The enthalpy of vaporization ($\Delta$Hvap) values from SIMPOL.1 calculations are presented in table B1 for each species considered in this work. These $\Delta$Hvap are always higher than 50 kJ/mol and they are in the range of 54 - 132 kJ/mol.

The sentence in page 14 lines 8-9 is modifed in the revised version of the paper as follows: « This is lower than the $\Delta$Hvap values calculated for individual components using SIMPOL.1. The calculated $\Delta$Hvap values are in the range of 54 - 132 kJ/mol. »

**Section 5.1: can the authors comment on how representative they believe woodfire stove smoke emissions are of wildfires?**

We do not believe that woodfire stove smoke emissions are representative of wildfires. For example, smog chamber experiments do not take into account all the different types of the burned vegetation. However, the identification of SOA precursors from smog chamber experiments of woodfire stove smoke emissions is an indication of which SOA precursors may be involved in wildfires. The following sentence is added page 15 line 6 : « Bruns et al. (2016) identified the most significant gaseous VOC precursors of SOA from residential wood combustion and presented their contribution to SOA concentrations. Although woodfire stove smoke emissions may not be representative of wildfires, they provide some indication of the SOA precursors involved during wildfires. »

**Section 5.3, lines 32-33 (first sentence of the section): would the left panel of Fig 7 technically be showing POAtot? Since these are the OAtot precursors?**

No, the text is correct, and Fig 7 is showing OAtot. OAtot precursors are made of POAtot and of the VOCs that are SOA precursors. In this study, POAtot corresponds to I/S/L-VOCs in the particle and gas phase and do not include VOCs.

Technically, Figure 7 presents all the OAtot precursors : POAtot (in the gas and

particle phase) represented in red and VOCs represented in blue.

**Section 6.1. Pg 21 lines 3-5: Can the authors briefly justify the choice of using Multstep-withVOCs for this figure?**

We choose to use Multistep-withVOCs for this figure because it is the simulation that represents the reference configuration and takes into account the added VOCs.

**Page 21 lines 9-10: from which model run(s) does this data come from?**

The data comes from Multistep-withVOCs run.

The sentence in page 21 lines 9-10 is modified as follows: « Figure 12 shows the distribution of the OA concentrations formed from the different VOCs emitted by wildfires in the simulation Multistep-withVOCs, over the sub-region MedReg during the summer 2007. »

**Pg 23, lines 10-13: how were the differences within "the fire plume" determined? What's meant by the fire plume here? How well can the model resolve an individual plume? Please explain this further.**

We mean here by fire plume, the panache of fire emissions transported far from the fire region. As our model is an eulerian model, we do not follow the fire plume, but its location is determined visually. The differences within the fire plume are calculated by considering the relative differences of PM2.5 concentrations between the simulations Multstep-withVOC and Multstep-unNMOG-withVOC.

The sentence in page 23 lines 10-13 are modified as follows : « Estimating the gaseous I/S/L-VOCs emissions from POA rather than from NMOG results in higher local PM2.5 concentrations (+8 to +16% in Greece) and lower PM2.5 concentrations mainly in Balkans (-30%) and in the fire plume visually determined (-8 to -16%). »

**Two more general comments:**

**A) Are the model results being compared to actual observations? If instead they are being compared to work done in the first author's other ACPD paper, this should be made more clear and the comparisons could be spelled out more explicitly.**

In this paper, the model results were not compared to observations. Comparisons to observations were performed in Majdi et al. (2019) (already published in ACP). They were not repeated here, because there was no observation of OA near the fire regions during the summer 2007.

The work done in Majdi et al. (2019) compared PM2.5 concentrations and optical

properties. Their reference simulation corresponds to the simulation onestepISVOC of this paper. According to Majdi et al. (2019), good general performances of the model are shown for the PM2.5 concentrations during the summer 2007. However, the 8 AIRBASE stations used for the evaluation of PM2.5 are far from the fire regions, and they may not provide meaningfull information for our study here.

Page 4, the sentence « Through comparisons to both ground based and satellite remote sensing (MODIS) observations, a general good performance for surface modeled PM2.5 with a clear improvement of PM2.5 is found when including fire emissions » is removed. It is replaced by the following sentence at the end of section 3 : «The reference simulation uses the same setup as Majdi et al. (2019). The evaluation of Majdi et al. (2019) of the simulation includes both ground based and satellite remote sensing (MODIS) observations. Ground-based observation of PM2.5 at 8 AIRBASE stations and of aerosol optical depth at 6 AERONET stations are used. The evaluation shows good performances of the model, especially when wildfires are taken into account in the simulation. Enhancements in PM concentrations due to wildfires are simulated at ±1-day uncertainty in the timing compared to satellite observations (MODIS), with a strong contribution from organic compounds (~61%) (Majdi et al., 2019).»

Page 13, line 11, the words « The simulation onestepISLVOC » are replaced by « The reference simulation onestepISLVOC ».

**B) This work would benefit from a discussion of the pros/cons of each model simulation type, and whether or not any model simulations appear to better represent the real atmosphere. Much work was clearly done here, but the paper currently does not seem to have the "why this matters/how it improves upon previous work" factor yet that will allow it to become an easily useful guide and reference for other members of the community.**

The conclusion was modified to better stress out how this work improves upon previous work, and how it can be useful for other members of the community.

« This study quantified the relative contribution of OAtot precursors (VOCs, I/S/L-VOCs) emitted by wildfires to OA formation and particle concentrations, during the summer 2007 over the Euro-Mediterranean region. A new chemical mechanism H2Oaro was developed to represent the SOA formation from selected VOCs, namely toluene, xylene, benzene, phenol, cresol, catechol, furan, guaiacol, syringol, naphthalene, methylnaphthalene, the structurally assigned and unassigned compounds with at least carbon atoms per molecule (USC>6), based on smog chamber experiments under low and high-NOx conditions. This mechanism was implemented in the chemistry transport model Polair3D of the air-quality platform Polyphemus. Over the Euro-Mediterranean area, the OA concentrations emitted by wildfires originate mostly from I/S/L-VOCs. The OA concentrations from gaseous I/S/L-VOCs are about 10 times higher than the OA concentrations from VOCs. However,

the contribution of the oxidation of VOCs to the OA concentrations is locally significant (it reaches 30% close to the area where wildfires are emitted and 20% in the fire plume). Air-quality models often represent SOA formation from only a few VOCs, such as toluene and xylene. This study points out the need to consider the contribution of a variety of VOCs, namely, phenol, benzene, catechol, cresol, xylene, toluene and syringol, when modelling SOA formation from wildfires. The contribution of these VOCs may even be underestimated here for two reasons. First, the yields from smoke chamber experiments were not corrected for wall losses, and they may therefore be underestimated leading to an underestimation of the SOA formation from VOCs in the model. Second, a large part of OA concentrations from VOCs is in the gas phase ( 70%). This suggests that the influence of the VOC emissions on OA concentrations could be larger, if the surrogates from these VOC oxidations partition more easily to the particle phase. This could be the case if further ageing mechanisms are considered for these VOCs or if the particles are very viscous (Kim et al., 2019). Emissions of gaseous I/S/L-VOCs are a large source of uncertainties. However, similar estimates were obtained here by using as a proxy POA emissions (with a factor of 1.5) or NMOG emissions (with a factor of 0.36). Sensitivity simulations were performed to quantify the uncertainties on OA and PM2.5 concentrations linked to I/S/L-VOCs emissions and chemical evolution (ageing). They are found to be lower than the uncertainties associated with SOA formation from VOC emissions. This stresses the need to consider a variety of VOCs in SOA formation model, and to better characterize their emission factors. »

**Figures/tables:**

**Figures 1-4 would benefit from being made in a higher-quality format rather than the excel default graphs.**

Figures 1-4 are reproduced in a high quality format in the revised version of the paper.

**Figure 11: It should state in the figure caption and/or on the figure which model simulation is being used to make this figure.**

The caption of figure 11 is modified in the revised version of the paper as follows: « Daily mean surface OA concentrations from wildfires (left panel) and the relative contribution of VOCs to OA from wildfires (right panel) during the summer 2007 (simulation Multistep-withVOCs). »

**Figure 12: from what data/model simulation(s) was this pie chart constructed? This should be stated in the figure caption and in the text.**

The caption of Figure 12 is modified in the revised version of the paper as follows:

« Distribution of OA concentrations formed from the different VOCs emitted by wildfires over the sub-region MedReg during the summer 2007 (simulation Multistep-withVOCs).»

**Figure 13: the colorbars should have units with them (% and mass concentrations?). This colorbar is a little hard to interpret, are we to take that the tan regions are anywhere between 0-5 or 8% different? Can the authors make the colorbars for each % difference plot the same, they're currently changing by between 5 and 11 units. I suggest considering a non-linear colorbar to see more structure within the -5 to 5% difference range.**

Units on the colorbars (% and $\mu g/m^3$) are added to Figure 13 in the new version of the revised paper.

Similar colorbar for each % difference plot are considered in Figure 13 in the new version of the revised paper.

**Technical comments:**

**Page 2 line 7: suggest rewriting to PM is composed of organic and inorganic compounds, dust, and black carbon (Jimenez et al., 2009).**

The sentence in page 2 line 7 is modified in the new version of the revised paper as follows: « PM is composed of organic and inorganic compounds, dust and black carbon (Jimenez et al., 2009).»

**Page 2 line 13: do the authors mean that both POA and SOA are composed of compounds of different volatilities? Suggest clarifying this sentence.**

Indeed, we mean that both POA and SOA are composed of compounds of different volatilities.

The sentence in page 2 line 13 is modified in the revised version of the paper as follows: « Both POA and SOA may be composed of components of different volatilities such as S-VOCs, L-VOCs which may partition between the gas and particle phases (Robinson et al., 2007). »

**Page 20 line 11: un should be Un in the simulation name.**

Multistep-unNMOG-withVOCs in page 20 line 11 is replaced by Multistep-UnNMOG-withVOCs (as in the simulation name) in the revised version of the paper.

**Page 20 line 21: this would make more sense if it was written something like "Across our cases, 28 to 42%..."**

The sentence in page 20 line 21 is modified in the revised version of the paper as follows: « Across our cases, 28 to 42% of the OA concentrations from I/S/L-VOCs emissions are primary. »

---

## Referee Report (RR1)

**Referee Report for acp-2018-1065**

**"Precursors and formation of secondary organic aerosols from wildfires in the Euro-Mediterranean region" by Majdi et al.**

The authors replied to reviewers' comments satisfactorily. The manuscript was revised according to the comments of referees.

I recommend the revised manuscript to be accepted as it is, with one technical suggestion:

Section 6 has subsections as 6.1 and 6.1.1. I think it would make more sense to replace 6.1.1 with 6.2.

---

## Author Response (AR2)

**Co-Editor Decision: Publish subject to technical corrections (01 Apr 2019)**
* * *
Dear co-editor,

We wish to thank you again for your corrections. All corrections have been included in this new version, considering the modification below.

Best regards,

 On behalf of authors Marwa Majdi

**1/ Page 3, line 10-11: Fragmentation and functionalization should be defined here (including the impact on volatility each process has).**

These sentences are added in page 3 line 11: "Fragmentation corresponds to the cleavage of C-C bounds, and it leads to oxidation products of lower carbon number and higher volatility than the precursor. Functionalization corresponds to the addition of oxygen-containing functional groups, and it leads to oxidation products of higher oxygen number. "

**2/ Page 4, first paragraph: Please clarify in the text whether the new H2O_aro scheme included the original H2O VOC precursors.**

The sentences  in page 4 line 1-4: "The objective of this work is to quantify the contribution of recently identified SOA precursors from wildfires (toluene, xylene, guaiacol, syringol, benzene, phenol, catechol, cresol, furan, naphthalene, methylnaphthalene and USC>6 compounds). To that end, a SOA formation mechanism is developed for those precursors, based on smog chamber experiments under low and high-NOx conditions."  are replaced by "The objective of this work is to quantify the contribution of recently identified SOA precursors from wildfires (guaiacol, syringol, benzene, phenol, catechol, cresol, furan, naphthalene, methylnaphthalene and USC>6 compounds). To that end, a new SOA formation mechanism is developed for those precursors, based on smog chamber experiments under low and high-NOx conditions. This new mechanism is used in conjunction with the $H^2O$ mechanism previously developed for biogenic and anthropogenic VOC precursors (xylene, toluene, isoprene, monoterpenes, sesquiterpenes...)."

**3/Page 5: Thank you for quantifying the error between the one-product model and the experimental data. Please define RMSE in the text; consider including the equation for RMSE in the text as well. (As shown in the author response to the original comment on this analysis).**

The equation for RMSE is included in the new version of the revised paper to define the RMSE.

The sentences in page 5 line 23-24: "The one-product model with a stoechiometric coefficient $\alpha 1$ of 0.28 and a vapor pressure of 4.59 10−8 torr correctly reproduces the experimental data wih a small amount of spread between the model and experimental data (RMSE of 3.1%). " are replaced by " The one-product model with a stoechiometric coefficient $\alpha 1$ of 0.28 and a vapor pressure of $4.59\ 10^{-8}$ torr correctly reproduces the experimental data. To quantify the spread  between the model and experimental data,  RMSE is used  as a statistical estimator and  calculated as:

RMSE (%) = 100 x ($\sum$ (Yieldexp-Yieldmodel)2 / N)1/2

where $Yield_{model}$ refers to the modeled SOA yield; $Yield_{exp}$ is the experimental SOA yield and N is the   number of experiments. A small amount of spread  between the model and experimental data (RMSE of 3.1%) is quantified."

**4/ Section 6 has subsections as 6.1 and 6.1.1, and it would make more sense to replace 6.1.1 with 6.2.**

Section 6.1.1 is replaced by section 6.2 in the new version of the revised paper.

[revised manuscript text omitted]